# Structural Similarities and Overlapping Activities among Dihydroflavonol 4-Reductase, Flavanone 4-Reductase, and Anthocyanidin Reductase Offer Metabolic Flexibility in the Flavonoid Pathway

**DOI:** 10.3390/ijms241813901

**Published:** 2023-09-09

**Authors:** Jacob A. Lewis, Bixia Zhang, Rishi Harza, Nathan Palmer, Gautam Sarath, Scott E. Sattler, Paul Twigg, Wilfred Vermerris, ChulHee Kang

**Affiliations:** 1Department of Chemistry, Washington State University, Pullman, WA 99164, USA; jacob.lewis2@wsu.edu (J.A.L.); bixia.zhang@wsu.edu (B.Z.);; 2Wheat, Sorghum, Forage Research Unit, U.S. Department of Agriculture—Agricultural Research Service, Lincoln, NE 68583, USA; nathan.palmer@usda.gov (N.P.); gautam.sarath@usda.gov (G.S.); scott.sattler@usda.gov (S.E.S.); 3Biology Department, University of Nebraska at Kearney, Kearney, NE 68849, USA; twiggp@unk.edu; 4Department of Microbiology & Cell Science and UF Genetics Institute, University of Florida, Gainesville, FL 32610, USA; wev@ufl.edu

**Keywords:** 3-deoxyanthocyanidin, flavonoid, *Panicum virgatum*, sorghum, *Sorghum bicolor*, switchgrass

## Abstract

Flavonoids are potent antioxidants that play a role in defense against pathogens, UV-radiation, and the detoxification of reactive oxygen species. Dihydroflavonol 4-reductase (DFR) and flavanone 4-reductase (FNR) reduce dihydroflavonols and flavanones, respectively, using NAD(P)H to produce flavan-(3)-4-(di)ols in flavonoid biosynthesis. Anthocyanidin reductase (ANR) reduces anthocyanidins to flavan-3-ols. In addition to their sequences, the 3D structures of recombinant DFR, FNR and ANR from sorghum and switchgrass showed a high level of similarity. The catalytic mechanism, substrate-specificity and key residues of three reductases were deduced from crystal structures, site-directed mutagenesis, molecular docking, kinetics, and thermodynamic ana-lyses. Although DFR displayed its highest activity against dihydroflavonols, it also showed activity against flavanones and anthocyanidins. It was inhibited by the flavonol quercetin and high concentrations of dihydroflavonols/flavonones. SbFNR1 and SbFNR2 did not show any activity against dihydroflavonols. However, SbFNR1 displayed activity against flavanones and ANR activity against two anthocyanidins, cyanidin and pelargonidin. Therefore, SbFNR1 and SbFNR2 could be specific ANR isozymes without delphinidin activity. Sorghum has high concentrations of 3-deoxyanthocyanidins in vivo, supporting the observed high activity of SbDFR against flavonols. Mining of expression data indicated substantial induction of these three reductase genes in both switchgrass and sorghum in response to biotic stress. Key signature sequences for proper DFR/ANR classification are proposed and could form the basis for future metabolic engineering of flavonoid metabolism.

## 1. Introduction

Flavonoids comprise a vast array of compounds with a three-ring structure C_6_(A)-C_3_(C)-C_6_(B) that vary in the chemical bonds of all three rings (Figure 1). These plant compounds play a role in nodulation, defense against pathogens and insects, attracting pollinators, and protection against UV-radiation, reactive oxygen species and cold temperatures [1,2]. In addition, the flavonoid tricin has been shown to be part of the lignin polymer in several monocot species [3,4,5], and naringenin was also demonstrated to become incorporate into the lignin of hybrid poplar plants overexpressing chalcone synthase 3 [6]. Modifying lignin composition through the incorporation of flavonoids offers opportunities to tailor cell wall properties. Flavonoids also serve as natural, non-toxic food colorants, and are of interest because of their potential medicinal benefits for humans. Even though it is challenging to demonstrate health benefits associated with the consumption of flavonoids at the level of individuals or populations, in vitro studies with flavonoids or flavonoid-derived metabolites in blood plasma have been shown to reduce inflammatory responses as they occur in cardiovascular disease [7] and obesity [8], which recently has been reviewed [9,10,11]. Additionally, traditional medicines, including “Dragon’s Blood” (*Daemonorops draco*) and the Asian shrubs *Fordia cauliflora* and *Millettia pulchra*, contain high levels of flavonoids, which could contribute to their health benefits [12,13,14].

Flavonoid production may have evolved in response to stress exposure during the transition from aquatic to terrestrial life. To date, over 6000 flavonoids have been identified and classified as flavan-3-ols, flavones, flavonols, and anthocyanidins, and isoflavones (Figure 1A) [15]. Anthocyanidins are responsible for flower petal color: pelargonidin causing orange, cyanidin resulting in red, and delphinidin producing purple flowers [16]. The majority of anthocyanidins are C3-hydroxylated. Further modifications can occur to anthocyanidins, such as glycosylation at their C3 and C5 positions, which results in a red shift. Further color shifts can occur by modifications of the glycosylated sites with either aromatic or acyl substitutions. Other derivatizations, such as aliphatic acylation, do not alter color, but increase solubility and stability [17]. The benefits of these molecules, such as anthocyanidin-3-*O*-glycosides, are well established with regard to their antioxidant activities [18]. In addition, anthocyanidins are characterized by a rather unstable oxonium ion formed at the O1 position on the C-ring through the activity of anthocyanidin synthase (ANS) [19].

Uniquely, 3-deoxyanthocyanidins (3-DOAs) that are derived from flavan-4-ols, lack the hydroxyl group on C-ring C3 (Figure 1) and, consequently, have no glycosylation or acylation substitutions in that position. This difference confers 3-DOA with higher stability to pH and temperature variations in comparison to their anthocyanidin analogs [20]. 3-DOA are potent antioxidants that also serve as phytoalexins that play an important role in the defense against pathogens, forming small oily droplets at the invasion site, and inhibiting pathogen growth [21]. Sorghum (*Sorghum bicolor*) is considered the only dietary source of 3-DOA, with concentrations in the seed of up to 10 mg/g [22,23].

Dihydroflavonol 4-reductase (DFR, EC1.1.1.219) catalyzes stereo-specifically the reduction in dihydroflavonols to flavan-3,4-diols, which is the rate-limiting step in anthocyanidin biosynthesis [24]. Three primary substrates of DFR have been identified, each differing by the number of hydroxyl sites on their B-ring (Figure 1B). Dihydrokaempferol (DHK) has a single hydroxyl group present on the B-ring at the C4′ position, whereas dihydroquercetin (DHQ) has an additional hydroxyl group at C3′, and dihydromyricetin (DHM) has three hydroxyl groups at C3′, C4′, and C5′ (Figure 1B). DFR converts these substrates to leucoanthocyanidin, leucopelargonidin, and leucodelphinidin, respectively. Availability of the substrates DHQ and DHM is dependent on catalysis by two cytochrome P450 hydroxylases, F3′H and F3′5′H, respectively. These P450 enzymes catalyze the hydroxylation of the B-ring of DHK and naringenin at either the C3′ position or the 3′ and 5′ positions (Figure 1A). Some plants lack flavonoid pigmentation due to the absence of F3′5′H activity and its product, DHM. Although these organisms lack DHM in vivo, their DFR is still able to convert the substrate to leucodelphinidin [17]. Substrate specificity of DFR has been studied across a wide variety of plants, as specific substrates are known to dictate flower and foliage color [25]. DFR of many species tends to reduce all three substrates, DHK, DHQ, and DHM. However, DFR from certain flowering plants such as petunia can only reduce DHK but are able to produce red flowers following a transformation with the *Zea mays DFR* gene [26]. In *Gerbera*, a missense mutation causing an asparagine- to-leucine substitution in the substrate-binding pocket of its DFR enabled the effective reduction of DHK to produce orange pigments, even though the wild-type enzyme was unable to reduce DHK [27]. DFR in rice was found to aide in the prevention of death from bacterial infection and overexpression of DFR increased production of NADPH [28]. Based on the structure of DFR from grape (*Vitis vinifera*), this enzyme belongs to the short-chain dehydrogenases/reductases (SDRs) [29,30], which is one of the largest NAD(P)H-dependent oxidoreductase families and contains a catalytic triad typically Ser-Lys-Tyr [29,30,31].

The activity of flavanone 4-reductase (FNR, EC:1.1.1.234) has been associated with the expression of *Sorghum bicolor* gene *Sobic.006G226800.1*, based on leaf color changes upon wounding. FNR produces flavan-4-ols, apiforol, and luteoforol, from the corresponding flavanones, naringenin and eriodictyol, respectively [32]. In some species, DFR also displays activity against eriodictyol and naringenin, which is probably due to their structural similarity to DHK and DHQ [33]. Although there is overlapping activity between FNR and DFR, DFR was determined to have lower activity for those substrates of FNR in sorghum [32]. In contrast, apple (*Malus domestica*) and pear (*Pyrus communis*) contain enzymes that could catalyze the reduction in both dihydroflavonols and flavanones [34].

In addition to overlapping activity between DFR and FNR, dihydroflavonols are the substrate of both flavonol synthase (FLS) and DFR, competing for these substrates and thus dictating whether flavan-3,4-diols or flavonols accumulate (Figure 1A). Therefore, metabolic flux associated with these two enzymes has been shown to be negatively correlated with each other [35]. In addition, two flavonols, quercetin and myricetin, the products of FLS have been shown to bind to DFR, which might inhibit its activity through binding to the active site [36]. Transcriptional regulation of directionality of the shunt between flavan-3,4-diols and flavonols has been identified in *A. thaliana*, whereas gene expression analyses indicate that other closely related genes, phenylalanine lyase (PAL), chalcone synthase (CHS), DFR, glutathione transferase (GST), chalcone isomerase (CHI), flavanone 3-hydroxylase (F3H) and FLS1 appear to be regulated in response to environmental factors [37]. In sorghum, biotic stresses are known to induce accumulation of 3-DOA through the induction of PAL, CHI, CHS, and DFR gene expression, while expression of F3H and ANS are downregulated [38].

Both sorghum and switchgrass (*Panicum virgatum*) are strategic biomass crops that can help support a sustainable bioeconomy in the U.S. and other countries. The value of these species will benefit from enhanced tolerance to biotic and abiotic stresses, modification of biomass composition to enhance industrial processing, and production of food-grade antioxidants in the seed. Tailoring the flavonoid biosynthetic pathway via genetic approaches offers prospects to accomplish this, but first requires a detailed understanding of substrate specificity, catalytic mechanisms, and inhibitors. So far, the critical residues dictating the substrate-specificity of DFR/FNR/ANR are poorly understood. In this study, we characterized those three critical reductases in the flavonoid pathway: DFR enzymes encoded by switchgrass gene *Pavir.5KG450100* (*PvDFRa*) and sorghum gene *Sobic.004G050200* (*SbDFR3*), as well as SbFNR1, SbFNR2, and SbANR encoded by *Sobic.006G226800.1*, *Sobic.006G226700.1,* and *Sobic.006G227200*, respectively. Our analysis reveals substrate preference, kinetic profiles and participating residues, inhibition patterns and overlapping activity among DFR, FNR, and ANR, which offers prospects for rerouting metabolic flux towards compounds of specific interest. 

## 2. Results

### 2.1. Determination of the Structure of DFR, FNR and ANR

The quality of diffraction data for various forms of PvDFRa, SbFNR1 and SbFNR2, and their respective statistics are listed in Table 1.

#### 2.1.1. PvDFRa

The crystal of purified PvDFRa enzyme belongs to a tetragonal space group, P42212 (PDBID: 8FEM) (Figure 2), with one DFR molecule in the asymmetric unit. Diffraction data at 2.3 Å resolution was collected from the Advanced Light Source (ALS, beam line 5.0.1) at a temperature of 100 K. The structure of PvDFRa was determined by molecular replacement using coordinates of *Vitis vinifera* DFR (PDBID: 2C29, VvDFR) that has the highest sequence similarity to PvDFRa in the Protein Data Bank (PDB). The space group of PvDFRa complexed with the substrate, DHQ (PDBID: 8FEN) was also tetragonal P42212 and the corresponding diffraction data was collected from ALS beam line 5.0.3 at 100 K with a resolution of 2.5 Å.

Analysis of the packing interface among individual PvDFRa molecules in the crystal lattice indicated little interaction among the symmetry-related molecules. Calculations through PDBePISA [39], which evaluates interactions between neighboring molecules in crystal lattices for the purpose of predicting biologically relevant oligomeric states, indicated that the Gibbs free energy of the binding interface (Δ^i^G) of 0.3 kcal·mol^−1^ between protein molecules, reflecting PvDFRa likely exists as a monomer in vivo. The structure of PvDFRa reveals thirteen α-helices and ten β-strands. Its Rossman fold domain, an NAD(P)H-binding signature, consisted of seven parallel β-strands (β1–β7) with both sides surrounded by eight α-helices (α1–α8). In particular, the existence of two α-helices (α7 and α8) between the β6 and β7 establish its overall topology of βαβαβαβαβαβααβα, closely resembling those of extended short-chain dehydrogenases/reductases (SDR).

In order to identify closely related 3D structural homologs in the PDB, a distance matrix alignment (DALI) search [40] that reports z-scores as a measure of similarity was performed. *Vitis vinifera* DFR (PDBID: 2IOD, VvDFR) was structurally most similar with a z-score of 48.9. Significantly, next was cinnamyl alcohol dehydrogenase 2 (CAD2) from *Medicago truncatula* (PDB ID: 4QTZ), which was reported to have DFR activity [41], having a z-score of 40.6. Cinnamoyl-CoA reductase (PDB ID: 5TQM) from *Sorghum bicolor* (SbCCR) had a z-score of 40.2, followed by vestitone reductase (PDB ID: 2P4H) from *Medicago sativa* and kavalactone reductase 1 (PDB ID: 6NBR) from *Piper methysticum*, both with a z-score of 39.9. The next proteins in the list include aldehyde reductase (PDB ID: 1UJM) and anthocyanidin reductase (PDB ID: 3FHS), but their z-scores are substantially lower. A BLAST search (Altschul et al., 1997) to identify proteins with similar amino acid sequences in the PDB revealed that VvDFR (PDB ID:2C29) again showed the highest identity (68%) to PvDFRa, followed by VvANR (PDB ID: 2RH8; 44%), vestitone reductase from *Medicago sativa* (PDB ID: 2P4H; 42%), CCR from *Petunia × hybrida* (PDB ID: 4R1S; 40%), CAD2 from *M. truncatula* (PDB identifier 4QTZ; 40%).

#### 2.1.2. SbFNR1 and SbFNR2

The binary complex crystal of SbFNR1 with NADP^+^ (PDBID: 8FET) and naringenin (PDBID: 8FEW) belongs to the tetragonal space group I4122. The ternary complex crystals of SbFNR1 with NADP(H) and naringenin were also obtained by soaking the naringenin binary complex crystals in mother liquor containing 10 mM NADPH (PDBID: 8FEU). SbFNR2 was also co-crystallized with naringenin in an orthorhombic space group P22121 (PDBID: 8FIP), and its ternary complex with naringenin and NADPH was obtained by soaking naringenin complex in other liquor solution containing 10 mM NADPH (PDBID: 8FIO). Diffraction data at 1.7 Å resolution were collected for both SbFNR1 and SbFNR2 crystals from ALS (beam line 5.0.1) at a temperature of 100 K and the structures of SbFNR1 and SbFNR2 were determined by molecular replacement using coordinates of the above PvDFRa. The PDBePISA calculation showed the Δ^i^G values are −2.3 and −1.6 kcal·mol^−1^ between the two molecules of SbFNR1 and SbFNR2 in the asymmetric unit, respectively, indicating a likelihood of existing as a monomer in vivo. Both SbFNR1 and SbFNR2 contained an increased number of secondary structures in contrast to PvDFRa with nineteen α-helices and eleven β-strands. As observed in PvDFRa, the Rossman fold of both FNRs consisted of seven parallel β-strands (β1–β7) with both sides surrounded by eight α-helices. In contrast to PvDFRa, there were two inserted peptides in the two SbFNRs, which were located between β5 and α5, α8 and α9 of PvDFRa (Figure 3). In addition, SbFNRs contained two extra beta strands, β1′ and β5′, which were located between β5 and α5, and α10 and α11, respectively.

A DALI search for SbFNR1 and SbFNR2 also indicated that VvDFR was the closest match (z-score of 41.4, 43.4), followed by *Petunia × hybrida* CCR (37.7, 39.2). The subsequent similar 3D structures were the same as those of PvDFRa: *Medicago truncatula* CAD2, *Sorghum bicolor* CCR, *Medicago sativa* vestitone reductase, *Piper methysticum* kavalactone reductase 1, aldehyde reductase and anthocyanidin reductase. A BLAST search for SbFNR1 and SbFNR2 showed a lower level of sequence identity with the same set of enzymes, and a slightly different order: VvANR (50%, 49%), followed by VvDFR (43%, 40%), vestitone reductase from Medicago sativa (36%, 35%) and CCR from *Petunia × hybrida* (36%, 37%).

#### 2.1.3. SbDFR3 and SbANR

Due to the high sequence similarity to PvDFRa, SbFNR1, and SbFNR2 (Figure 3), homology-modeling for SbDFR3 and SbANR was performed utilizing the Swiss-Model server [42] (Appendix A). As expected, the overall structures of both SbDFR3 and SbANR are highly similar to PvDFRa, SbFNR1, and SbFNR2 containing 18 α-helices and twelve β-strands with root mean square deviation (RMSD) values displayed in Appendix A.

### 2.2. Detailed Analysis of the Cofactor-Binding Pocket

From the early stages of refinement, a difference (*Fo–Fc*) electron density map clearly indicated the presence of an NADP^+^ molecule in the binding pocket of PvDFRa, SbFNR1, and SbFNR2 with its nicotinamide moiety properly positioned for the pro-*R* hydride transfer towards the catalytic site (Appendix A). The bound nicotinamide ring in PvDFRa, SbFNR1, and SbFNR2 established a *syn*-conformation, whereas the adenine ring adopted an *anti*-conformation (Appendix A). Despite a lack of cofactor in the crystallization buffer, an NADP^+^ molecule was present in the crystal structure of PvDFRa. However, the crystalized SbFNRs did not display any NADP^+^, thus had to be added for co-crystallization, which indicated that PvDFRa had a higher affinity for NADP^+^ than SbFNRs, as confirmed by our ITC results (Table 2). Having both apo- and NADP(H)-bound forms allowed us to observe the conformational change upon binding NADP(H) in the SbFNRs. Compared to the structure without NADP(H), the loop region, ^89^Val----Pro^100^ in SbFNR1 (and ^89^Val----Lys^102^ in SbFNR2) went through large conformational changes upon binding of NADP(H). Coincidentally, in the structure of VvANR, of which the structure is very similar to SbFNRs with an RMSD value of 0.83 Å, the corresponding loop is largely disordered in the absence of NADP(H) [30]. In FNR, DFR, and ANR (Figure 3), Arg-43, Lys-50, Asp-69, Thr-131, Lys-177, and Ser-215 were key residues for cofactor-binding and are completely conserved. Similar to other typical extended SDRs, the conserved motif, GXXGXXG, which is known to participate in the binding of the pyrophosphate group of NAD(P)H through a helical dipole of α1 [43], is observed at the first β-α-β unit in all compared enzymes (Figure 3). In addition, the β2 area of PvDFRa, SbFNR1, and SbFNR2 began with a GYXV motif, which differs from similar animal enzymes that do not have a conserved residue in the second position [44]. Preference for NADP(H) to NAD(H) is thought to be established by two positively charged residues in SDRs located at positions 29 and 52 (following the numbering system of PvDFRa). Among the compared SDRs, the residue at position 52 is conserved as arginine in all sequences, but position 29 is not conserved. NADP(H)-exclusive SDRs have a basic residue at position 29, which stabilizes the 2′ ribosyl phosphate of NADPH, indicating that both NAD(H) and NADP(H) could be utilized in all compared enzymes including PvDFRs, SbFNRs, and ANRs.

Detailed Analysis of the Substrate-Binding Pocket: To understand the observed substrate preferences and their overlapping activities among PvDFRa, SbDFR1, and SbDFR2, the substrate-binding pockets were examined for specific interactions between each enzyme and their associated substrates.

#### 2.2.1. DHQ in PvDFRa (PDBID: 8FEN)

A deep groove containing the substrate-binding pocket of PvDFRa was formed by three loops that connect β4 and α4, β5 and α5, β6 and α6, which was lined mainly with hydrophobic residues to reflect the non-polar nature of its substrate. In the ternary complex crystal structure, the C4 atom of NADP^+^ was 2.9Å away from the C-ring carbonyl C4 atom of DHQ where hydride transfer happens (Figure 4D). Although the positions and orientations of the catalytic triad, Thr-143, Tyr-178, and Lys-182, resembled those observed in other SDR-type enzymes, PvDFRa uniquely had Thr-143 instead of a serine residue typical among SDRs (Figure 3). The sidechain of this Thr-143 was positioned to act as a potential hydrogen bond donor for the C4 carbonyl oxygen with a distance of 3.02 Å. The sidechain of Tyr-178 was hydrogen bonded to the O2′ atom of nicotinamide ribose with a 2.7 Å distance. The sidechain of Lys-182 was also within a hydrogen bond interaction with both O2′ and O3′ of the nicotinamide ribose moiety at a distance of 3.19 Å and 2.83 Å, respectively. In addition, the sidechain of Asn-148 established a hydrogen bond interaction with the C3′ hydroxyl group on the B-ring at a distance of 2.33 Å. The side chain of Asn-148 was 2.83 Å from the C4′ hydroxyl group on the DHQ of the B-ring. Gln-242 stabilized the C4′ hydroxyl group via a hydrogen bond interaction being 2.71 Å away.

#### 2.2.2. Naringenin/DHQ in SbFNR1 (PDBID: 8FEU, 8FEV)

The substrate-binding pocket of SbFNR1 was also surrounded by hydrophobic residues located on a loop between β4 and α4, the 3_10_ helix between β5 and α5, and two α-helices of α7 and α8. Notably, the α8 in SbFNR1 caused a tighter connection between α7 and β3′ (Figure 4A,B), resulting in a narrower opening in SbFNR1 than that of PvDFRa, potentially reducing the rate of diffusion. There were four hydrophobic residues in α8 of SbFNR1 closely located to the bound substrates, while there was only one residue (Ile-237) in α8 of PvDFRa. In addition, the loop between β4 and α4 in SbFNR1 is located at a closer proximity to the bound substrate than PvDFRa. The A- and C-rings of the bound naringenin were located at a stacked position with the nicotinamide ring, resulting in a distance between the carbonyl C4 and nicotinamide C4 of 4.2 Å (Figure 4A). This carbonyl C4 on the C-ring of naringenin is hydrogen bonded to the sidechains of Ser-133 and His-173, both of which constitute the catalytic triad. The hydroxyl group on the B-ring formed hydrogen bonds with the sidechains of Gln-244 and Ser-240. The hydroxyl group on C7 of the A-ring was hydrogen bonded to the side chain and backbone amide of Asp-218 through a water molecule, and the hydroxyl group on C5 was bonded to the ribosyl hydroxyl group of NADPH. Due to the resolution of the ternary complex structure of FNR1 with naringenin and NADPH, it is impossible to tell if a reaction took place in the crystal lattice.

DHQ in the ternary complex of SbFNR1 with NADP^+^ was oriented with its B-ring in a stacking position with the nicotinamide ring and its A- and C-rings in opposite directions compared to that of bound naringenin (Figure 4B). The hydroxyl group on C7 of the DHQ A-ring was hydrogen bonded with the side chain and backbone amide of Gln-244. The carbonyl and hydroxyl groups on the C-ring were also hydrogen bonded with a water molecule that was connected to Tyr-138, and the hydroxyl group on C3′ on the B-ring was hydrogen bonded with ribosyl hydroxy group of NADP^+^. However, the C4 carbonyl of DHQ was 6.65 Å from the nicotinamide C4 atom, which is too far for proper hydride transfer and is consistent with a lack of activity of SbFNRs against DHQ (Figure 4B).

#### 2.2.3. Naringenin in SbFNR2 (PDBID: 8FIO)

The position and orientation of bound naringenin in the ternary complex structure of SbFNR2 did not favor its catalytic reaction. The C4 atom of the naringenin carbonyl group was 5 Å away from C4 of NADP^+^ and its oxygen was not hydrogen bonded to any of the catalytic triad residues, which is consistent with the lack of activity observed with this substrate in SbFNR2 (Figure 4C). The *para*-hydroxyl group on the B-ring of naringenin was hydrogen-bonded to the phenolic side chain of Tyr-138 and the backbone amide of Leu-200. The same hydroxyl group was also connected to the backbone and side chain of Gln-242 through a water molecule. The hydroxyl group on C5 of the A-ring established a direct hydrogen bond with the side chain of Thr-212 and an indirect hydrogen bond with the pyrophosphate group of NADPH through a water molecule. C7 of the A-ring was also hydrogen bonded with the side chain of Asp-216 and the backbone amide of Thr-212.

### 2.3. Molecular Docking to Investigate the Interactions between PvDFRa and Substrates

Despite numerous attempts for both soaking and co-crystallization of DHM and DHK, we were not able to obtain a reliable electron density for those compounds in the crystal structure of PvDFRa. Thus, molecular docking was performed for all five compounds (DHQ, DHM, DHK, naringenin, eriodictyol) using Autodock Vina ([45,46] (Appendix A)). The docking energy, ∆G_binding_, of the tested substrates, were −8.821, −8.661, −8.321, −7.971, and −7.701 kcal·mol^−1^ for DHQ, naringenin, DHM, eriodictyol, and DHK, respectively. Supporting the molecular docking results, the docked DHQ position is nearly identical to that observed in the DHQ ternary complex crystal structure of PvDFRa supporting the accuracy of docking. In contrast, the B-ring of the docked DHK molecule displayed a stacking interaction with a phenyl side chain of Phe-179 and its lone hydroxyl group established a hydrogen bond with a phenyl side chain of Tyr-174. Significantly, the B-ring of DHK adopted a large bend, which was almost perpendicular to its A and C-rings. However, DHQ and DHM did not show any distorted B-ring contortion as observed with DHK, and the hydroxyl groups on the B-ring of both DHQ and DHM were within hydrogen bond distance from the sidechains of Asn-148 and Gln-242. Both eriodictyol and naringenin bound at the active site also adopted a bent conformation but in the opposite direction to that of DHK’s B-ring (Appendix A). The hydroxyl groups on the B-ring of naringenin established hydrogen bond interactions with both the nitrogen and the oxygen of the sidechain of Gln-242.

### 2.4. Enzyme Activity Assays of PvDFRa, SbDFR3, SbFNR1, SbFNR2 and SbANR

To investigate the enzyme kinetics and substrate specificity for DFR, FNR, and ANR, enzyme activity assays were conducted by following similar protocols used for DFR from tea (*Camellina sinensis*) [47], with minor changes.

#### 2.4.1. Activity against Dihydroflavonol (DHQ, DHM and DHK)

The *K_m_* values for PvDFRa were 115.4 µM^−1^ for NADPH and 190.6 µM^−1^ for NADH (Table 3, Appendix A). *k_cat_* was 3.906 min^−1^ and 3.095 min^−1^ for NADPH and NADH, respectively, indicating a slight favor of PvDFRa for NADPH over NADH as shown in Table 3 and Appendix A. The *K_m_* value of PvDFRa was 191.2 µM for DHK, 150.0 µM for DHQ, and 128.6 µM for DHM. The *k_cat_* values for DHK, DHQ, and DHM were 0.05737 min^−1^, 12.13 min^−1^, and 1.986 min^−1^ respectively. As shown in Figure 5, inhibition was observed in PvDFRa at higher concentrations of DHQ and DHM. However, the *K_I_* was much greater than the *K_m_*, hence could not be quantified.

PvDFRa has a unique catalytic residue, Thr-143, instead of serine as observed in other SDRs. The *K_m_* value of PvDFRa T143S mutant was 19.04 µM for DHQ, 7.409 µM for DHM, and 137.7 µM for DHK (Table 3), displaying an increased catalytic efficiency 1.09-fold for DHK, 3.53-fold for DHQ and 2.26-fold for DHM. However, the preference for substrates was not impacted by the mutation T143S, indicating its unimportance in substrate specificity. On the other hand, the *K_m_* value of Q242A mutant PvDFRa was 156.8 µM for DHQ, 119.2 µM for DHM, and 74.59 µM for DHK (Table 3). The *k_cat_* values were 2.571 min^−1^ for DHQ, 2.273 min^−1^ for DHM, and 0.4701 min^−1^ for DHK, displaying a higher catalytic efficiency for the least polar B-ring substrates amongst the three, DHK, by 1.7-fold.

On the other hand, the *K_m_* values for SbDFR3 were 166.4 µM^−1^ for DHQ, 118.5 µM^−1^ for DHM, and 117.6 µM^−1^ for DHK (Table 3). The *k_cat_* values for SbDFR3 were 3.795 min^−1^, 4.640 min^−1^, and 0.608 min^−1^ for DHQ, DHM, and DHK, respectively. Both *K_m_* and *k_cat_* indicated that DHM is the most preferred substrate for SbDFR3, contrary to PvDFRa’s favored DHQ.

SbFNR1, SbFNR2, and SbANR displayed no significant activity against DHM, DHK, and DHQ indicating there is no overlapping activity with DFR for those dihydroflavonols.

#### 2.4.2. Activity against Flavanones (Eriodictyol and Naringenin)

To measure any overlapping specificity between FNR and DFR, the ability of PvDFRa, SbDFR3, SbFNR1, and SbFNR2 to reduce flavanones such as eriodictyol and naringenin was assayed using similar conditions to those used for dihydroflavonols. The products of the enzymatic reactions with both eriodictyol and naringenin were confirmed by MALDI-MS (Appendix A). For PvDFRa, the *k_cat_* values for eriodictyol and naringenin were 0.8481 min^−1^ and 3.549 min^−1^, respectively. The *K_m_* of PvDFRa for eriodictyol and naringenin were 246.5 µM^−1^ and 407.6 µM^−1^, respectively. Again, substrate inhibition occurred at higher concentrations of those flavanones. The *k_cat_* values for eriodictyol and naringenin were 0.1236 min^−1^ and 2.084 min^−1^, respectively, for PvDFRa. The *K_m_* for eriodictyol and naringenin were 25.21 µM^−1^ and 118.2 µM^−1^, respectively, for SbDFR3.

Although the activity of SbFNR1 was observed again with both naringenin and erio-dictyol in our assay, a much longer incubation time was required to obtain a detectible amount of the corresponding products, apiforol and luteoforol. Thus, a meaningful kinetic assay measuring the initial velocities could not be performed properly. Compared to PvDFRa and SbDFR3, SbFNR1 displayed only ~3% product yield (Figure 6). SbFNR2 and SbANR did not display any activity against naringenin and eriodictyol despite substantially longer reaction times.

#### 2.4.3. Activity against Anthocyanidins (Cyanidin, Pelargonidin and Delphinidin)

To measure any overlapping specificity among FNR, DFR, and ANR for anthocyanidins, their activity against cyanidin, pelargonidin, and delphinidin were assayed. Both SbFNR1 and SbFNR2 displayed substantial ANR activity. Cyanidin was reduced to its product by both SbFNR1 and SbFNR2 with 11.9% and 7.4% efficiency, respectively, relative to the activity of SbANR (Appendix A). Pelargonidin was also reduced to the resulting product by SbFNR1 and SbFNR2 with 2.1% and 42.7% efficiency, respectively, relative to SbANR. However, both SbDFR3 and PvDFRa showed marginal activity against cyanidin, with 0.9% and 0.5% of the activity of SbANR, respectively (Figure 6). For pelargonidin, SbDFR3 and PvDFRa displayed 2.3% and 13.8% activity, respectively (Figure 6).

LC-MS data indicated that multiple peaks arose from the use of anthocyanidin as a substrate in both assays of SbFNR1 and SbFNR2 (Appendix A). This is congruent with previous reports identifying that the enzymatic reaction of ANR with anthocyanidin yields non-stereospecific products [30,48,49]. Four peaks produced by SbFNR2 against cyanidin were identified as four stereoisomers (Appendix A): l-catechin (2S,3R), d-catechin (2R, 3S), l-epicatechin (2R, 3R), and d-epicatechin (2S,3S) (*m*/*z* = 291.0900). To probe the possible binding sites and conformation of cyanidin to SbFNR2, cyanidin was docked into its crystal structure and identified two possible orientations, each for hydride transfer to either C2 atom or C4 atom of the C-ring (Figure 7A,B). The four possible intermediates after the first hydride transfer were also docked to SbFNR2 to identify the stereo-specific binding position and orientation for each molecule. Those intermediates (Figure 7C–F) have binding affinities of −8.874 kcal·mol^−1^, −8.547 kcal·mol^−1^, −9.181 kcal·mol^−1^, and −8.242 kcal·mol^−1^, which will become d-catechin, l-epicatechin, d-epicatechin and, l-catechin, respectively.

### 2.5. Inhibition by Quercetin

Our assay of PvDFRa with the flavonol quercetin indicated that quercetin is not a substrate (Appendix A). It acts as an inhibitor of DFR activity (Figure 5E). To confirm it, an enzyme activity assay of PvDFRa was conducted utilizing DHM as the substrate with varying concentrations of quercetin as an inhibitor (Figure 5E). As observed previously, product-inhibition was observed at higher concentrations. The *K_m_* for DHM remained the same, 100.5 μM, with all tested concentrations of quercetin, and *V_max_* decreased with increasing concentrations of quercetin, from 5.427 min^−1^ in the absence of quercetin to 4.193 min^−1^ and 3.545 min^−1^ in the presence of 75 μM and 150 μM quercitin, respectively. The unaffected *K_m_* and decreased *V_max_* indicated the inhibition of PvDFRa by quercetin is a noncompetitive inhibition as observed in DFR from *Zanthoxylum bungeanum* at higher concentrations [50].

### 2.6. Isothermal Titration Calorimetry

In order to obtain thermodynamic parameters for the association, isothermal titration calorimetry (ITC) for PvDFRa was performed (Appendix A, Table 2). Consistent with *K_m_* values in the enzyme kinetic assay, PvDFRa displayed the same trend of dissociation constants (*K_d_*). The dissociation constants for NADP^+^, DHK, DHQ, DHM, naringenin, and eriodictyol were 184.6 nM, 18.65 µM, 16.87 µM, 21.23 µM, 176.9 µM, and 64.94 µM, res-pectively. The binding of the substrates and NADP^+^ was driven enthalpically. While the association of DHQ, DHM, eriodictyol, and NADP^+^ was opposed entropically, DHK and naringenin displayed a positive ΔS. These observations suggest that large confirmational changes to the active site are necessary for the binding of the NADP^+^, DHM, DHQ, and eriodictyol, but are not required for the association of DHK and naringenin (Table 2). Thermodynamic parameters of SbFNR1 were also assayed with ITC for association with NADP^+^, and it had a *K_d_* of 13.91 µM that indicated a 10-fold decrease in affinity for NADP^+^ compared to that of PvDFRa as predicted from the crystal structures.

## 3. Discussion

Many oxidoreductases involved in the plant secondary metabolism routes (terpenoids, alkaloids, phenolics) belong to the SDR superfamily. Despite their conserved ‘Rossmann-fold’ structure, those SDR enzymes exhibit low sequence similarities, which constituted a bottleneck in terms of identification and functional annotations of plant SDRs remain scarce [51]. Our comprehensive investigation of PvDFRa, SbDFR3, SbFNR1, SbFNR2, and SbANR indicated that three reductases in the flavonoid pathway share very similar structure of SDRs and a substantial overlapping substrate-specificity, which could be useful in classification.

### 3.1. Significance of Overlapping Substrate-Specificity among FNR, DFR and ANR

Judging from the abortive DHQ-complex structure of SbFNR1 (Figure 4B), its lack of DFR activity for dihydroflavonols is rather obvious. However, although the activity was much lower than those of PvDFRa and SbDFR3, SbFNR1 showed FNR activity towards flavanones, such as eriodictyol and naringenin reducing them to the corresponding flavan-4-ols, apiforol and luteoforol, (Figure 6). In addition, both SbFNR1 and SbFNR2 displayed ANR activity, reducing cyanidin and pelargonidin to epicatechin and epiafzelchin, respectively (Figure 6 and Appendix A). It is especially noteworthy that the activity of SbFNR2 against epiafzelchin reaches 43% of SbANR activity. However, similar to SbDFR3, neither SbFNR1 nor SbFNR2 displayed any activity against delphinidin and only SbANR showed activity to this anthocyanidin (Figure 6). Therefore, considering its marginal activity against naringenin and eriodictyol, the enzymes encoded by *Sobic.006G226800.1* (SbFNR1) and *Sobic.006G226700.1* (SbFNR2) might be special isoforms of ANR that lack activity against delphinidin.

In addition to an activity resemblance of SbFNRs to ANR, there are also noticeable structural similarities between SbFNRs and ANRs, which separates those reductases from DFRs (Figure 2). In SbFNR1, SbFNR2, SbANR, and VvANR, there are two sequence motifs at the sidewall of the substrate-binding pocket that are elongated relative to the corresponding motifs in DFR. Those are ^143^LLGDGHGH^150^/^234^IQKTSGE^242^ in SbFNR1, and ^133^LQGDGH^138^/^224^IETTSGA^230^ in SbFNR2, which comprise portions of the β1′ and α8 motifs, respectively. The substrate-binding pocket of FNR and ANR are also surrounded with negatively charged residues (Appendix A), which are Asp-226, Glu-227 in SbFNR1 and are conserved in both SbFNR2 and SbANR. Thus, electrostatic interaction with the oxonium ion of the anthocyanidin could allow its unique consecutive hydride transfer to multiple stereoisomers and their intermediates as discussed later. Overall, these features could be used as a motif distinguishing ANR/FNR from DFR despite their otherwise similar amino acid sequence, structure, and overlapping substrate specificities.

To further understand the relationship among DFR, ANR, and FNR, a phylogenetic tree was built (Figure 8). First, a BLAST search was conducted with PvDFRa exclusively for non-redundant protein sequences in *Sorghum bicolor*, *Panicum virgatum*, *Arabidopsis thaliana*, and *Oryza sativa*, and then followed by alignment and construction of a phylogenetic tree with Molecular Evolutionary Genetics Analysis (MEGA11). A clear separation is evident between DFR and ANR sequences, further separating AtDFR and AtANR from the three monocot DFRs and ANRs. Supporting our hypothesis, SbFNR1 and SbFNR2 were most similar to one of the SbANRs, XO 021318468. In addition, phylogenetic analysis suggests that those reductases in the flavonoid pathway might have evolved from the same or a similar ancestral short-chain dehydrogenase such as CCR (Figure 8).

In the flavonoid pathway, flavanones and dihydroflavonols could be in direct competition for the active site of PvDFRa. If naringenin accumulates due to reduced expression/activity of the P450 enzymes, F3′H, F3′5′H and F3H (Figure 1A), the concentration of DHK, DHQ, and DHM could be compromised due to this overlapping activity, resulting in reduced amounts of the corresponding flavonon-3-4-diols, leucoanthocyanidinin, leucopelargonidin and leucodelphinidin, but increased production of apiforol through combined activities of DFR and FNR (Figure 1A). A similar scenario is possible for eriodictyol based on relative activities between F3′H and F3H, which affect the luteoforol concentration. Reduced expression/activity of DFR could promote substrate channeling through the activity of FLS towards flavonols (Figure 1A). The overlapping activities between DFR and ANR suggest that the divergence among these enzymes in the flavonoid pathway enabled the efficient reduction in specific substrates necessary for that plant’s survival depending on stressors. For example, the increased level of flavan-4-ols in sorghum has been associated with increased fungal pathogen resistance [53]. In an instance when there is a high concentration of dihydroflavonols, the metabolic flux could be shifted to flavan-4-ols responding to the altered environmental conditions. Indeed, the in vivo regulation of DFR appears to be quite complex, which could interlink FLS, FNR, and ANR, as their active sites can accommodate multiple substrates that include flavaonols, dihydroflavonols, and flavanones [35]. Furthermore, flavonols such as quercetin act as inhibitors of PvDFRa (Figure 5E).

### 3.2. Broad Substrate-Specificity and Key Residues of DFR

The substrate specificity of DFR has been of great interest, particularly for potential engineering to modify flower colors [54]. Substrate-specificity can differ even among isozymes in the same species as observed in several plants. Switchgrass has two DFR isozymes, PvDFRa and PvDFRb, and sorghum has four isozymes, SbDFR1, SbDFR2, SbDFR3 and SbDFR4 displaying ~87% amino acid sequence identity among them (Appendix A). Our kinetic data indicated that PvDFRa prefers DHQ as a substrate over the other tested compounds, DHM, DHK, eriodictyol and naringenin in descending order in their *K_m_* and *K_m_*/*k*_cat_ values (Table 3). Both PvDFRa and SbDFR3 also displayed ANR activity against cyanidin and pelargonidin, but not delphinidin (Figure 5F and Appendix A). The dissociation constant (*K_d_*) from the ITC data agreed well with the hierarchical order for those substrates measured by enzyme kinetic assays suggesting that the binding affinity of DHK (26.42 µM) is lower than for DHQ (16.87 µM) and DHM (21.23 µM). (Table 2). Significantly, our molecular docking result suggests that the binding positions of DHK, eriodictyol, and naringenin were shifted away from those of DHQ and DHM adopting a similarly distorted torsional angle. This observation also agrees with the overall lower catalytic efficiency of PvDFRa for DHK, naringenin, and eriodictyol (Figure 1A, Table 3). On the other hand, SbDFR3 established a substantially reduced *K_m_*, thus higher affinity for substrates with fewer hydroxyl groups (naringenin, DHK, and eriodictyol) compared to that of PvDFRa. Therefore, enhanced expression of DFR in sorghum due to stress could increase the pool of flavan-4-ols and eventually 3-DOA.

Based on these substrate preferences, the substrate-binding pocket was examined to pinpoint specific interactions between PvDFRa and those associated substrates, which could be used for future engineering for DFR of different specificity. The first key difference determining substrate specificity seems to be the amino acid facing the B-ring, that is, Asn-148 (PvDFRa) and Ser-138 (SbDFR3). In the DHQ complex structure of PvDFRa, the sidechain of Asn-148 displayed a hydrogen bond interaction with a *p*-hydroxyl group in the B-ring of DHQ as in the VvDFR (Figure 4D). The corresponding residue to the Asn-148 in PvDFRa is conserved in VvDFR, (Asn-133), *Gerbera* DFR (Asn-134), and *Cymbidium* DFR (Asn-135), but in *Petunia* DFR as Asp-143. The kinetic data of SbDFR3, which contains a unique Ser-148 instead of Asn-148 in the other three SbDFR isozymes, displayed higher catalytic efficiency for DHM than DHQ (Table 3). Secondly, compared to DFR from other species, PvDFRa has a unique catalytic residue, Thr-143 instead of serine that is conserved in DFRs in most species. A mutant PvDFRa T143S displayed an increased catalytic efficiency for DHQ, DHK, and DHM without affecting the substrate specificity. Finally, abolishing the observed hydrogen bond interaction between the sidechain of Gln-242 and C4′ hydroxyl group of DHQ through a mutation Q242A in PvDFR favors the binding of the less polar substrates: naringenin eriodictyol, and DHK. Those three residues separately or in combination provide varied substrate-affinities contributing to strategic manipulation possibilities in sorghum and switchgrass.

### 3.3. Inhibition by NADP^+^, Anthocyanidins and Quercetin

In our experiment, the NADP^+^-binding affinity of PvDFRa was roughly 100 times greater than its affinity for DHQ (Table 2), and the structure of PvDFRa contained NADP^+^ in its active site even without added NADP^+^ in the crystallization buffer, which was not the case for the purified SbFNR1 and SbFNR2. These facts support the previous hypothesis that a non-productive NADP^+^ binary complex could limit the ability of the enzyme to replace NADP^+^ with NADPH [55] as an abortive DFR complex with NADP^+^ or a dihydroflavonol can inhibit the association of NADPH. The higher affinity of NADPH can inhibit the formation of those abortive complexes and thus, diminished concentration of NADPH could significantly limit DFR activity. In PvDFRa’s kinetic curves, substrate inhibition was observed as substrate concentration increased past the *V_max_* of DHK, DHQ, and DHM (Figure 5A), which could suggest a formation of inhibitory NADP^+^ complex by association of the substrate or undissociated product before NADPH binding. PvDFRa and SbDFR3 were able to accept both NADH and NADPH as the coenzyme for hydride transfer. However, when NADH is utilized instead of NADPH, enzyme efficiency is reduced to 60.55% of NADPH (Appendix A).

Previously, quercetin, a flavonol, complexes of VvDFR have been observed, where the formation of an NADPH-quercetin ternary complex led to inhibition of the DHQ reduction and there even establishes an inhibition complex containing two quercetin molecules in the active site [36]. Furthermore, recent studies of Chinese prickly ash (*Zanthoxylum bungeanum*) DFR identified the binding of two flavonols to the active site through ITC [50]. Our result indicated that quercetin was also an inhibitor of PvDFRa. Although the electron density of our quercetin complex structure of PvDFRa was not clear enough to show the exact interaction of two quercetin molecules in the active site, the disordered nature of associated quercetin could reflect its less specific interaction compared to dihydroflavonols or even flavanones. Significantly, flavonols such as quercetin, the product of FLS, are the most common flavonoids present in plant vacuoles [56]. Aside from providing defense against UV-B radiation, they facilitate the regulation of symbiosis, phytohormones, auxins, and male fertility in maize [57]. Carefully engineered DFR could increase flux towards flavonols to create colorless flowers due to the loss of pigmentation as well as the valorization of flavonols for medications. In fact, knockout studies of DFR in purple sweet potatoes identified the additional accumulation of quercetin-3-*O*-hexoside and quercetin-3-*O*-glucoside [58]. This hypothesis is congruent with our data suggesting the inhibition of PvDFRa activity by quercetin is mixed. Due to the structural similarity among the C_6_-C_3_-C_6_ flavonoids, it is tempting to speculate that reductases in the flavonoid pathway have an affinity for various flavonoid molecules that behave as competitive or mixed inhibitors regulating the overall pathway.

### 3.4. Catalytic Mechanism of DFR, FNR and ANR

DFR activity: Based on our crystal structures, kinetics assays, and ITC results, the catalytic mechanisms for DFR, FNR, and ANR are proposed (Appendix A). NADPH and the substrate associate into their binding pockets, located nearest the N- and C-terminal domains, respectively. For a productive reaction, NAD(P)H likely binds first followed by the substrate, as in other NAD(P)(H)-dependent oxidoreductases [59,60]. Considering the high affinity of PvDFRa for NADP^+^ (Table 2, Appendix A) and the observed bound flavone in the NADP^+^-binding pocket of SbFNR (PDBID: 8FEW), this order does not appear to be maintained well and tends to form an abortive enzyme complex. Binding of NADPH to the apo-form enzyme with its B-face of the nicotinamide ring open to the cleft requires a substantial change in its conformation. Several bound water molecules in the corresponding pocket of apo-form enzyme (PDBID: 8FEM, 8FEW) were replaced, indicating an entropic contribution for the association of cofactor. Then, Tyr-178 and Lys-182 establish a hydrogen bond network with the 2′ and 3′ hydroxyl groups of the associated NAD(P)H ribosyl ring, fixing the position of the nicotinamide and potentially lowering the *pK_a_* of the Tyr-178 as shown in other enzymes [61]. Lys-182 also forms a hydrogen bond between the sidechain of Tyr-178 and the O2′ hydroxyl group on the nicotinamide ribose would decrease the *pK_a_* of the tyrosine, allowing an effective proton transfer. Upon substrate binding, Thr-143, which is serine in other compared reductases, also establishes the hydrogen bond with the carbonyl oxygen of DHQ, which is located at 2.6 Å from the phenolate group of Tyr-178. In addition, the carbonyl carbon of bound DHQ is located at 2.8 Å from the C4 atom of nicotinamide ring of NADPH (Figure 4D). In the first catalytic step, pro-*R* hydride transfer occurs from the C4 atom of NADPH to the C4 atom of DHQ. The resulting oxyanion of the C4 carbonyl oxygen atom is temporarily stabilized by the oxyanion hole established from the side chains of Thr-143 and Tyr-182 (Appendix A). The collapse of the tetrahedral intermediate is then followed by the protonation of DHQ through Tyr-178. Thus, Tyr-178 serves as a general acid in substrate protonation. Together Tyr-178 and Lys-182 are able to establish a proton network protonating the C4 carbonyl of DHQ.

### 3.5. FNR Activity

As expected, neither SbFNR1 nor SbFNR2 displayed any activity against DHQ, DHM and DHK. They showed only marginal activity against naringenin and eriodictyol, which could be due to a significant formation of a non-productive complex (Figure 6). In SbFNR1 and SbFNR2, the corresponding catalytic triad is composed of Ser-133/His-173/Lys-177 and Ser-133/Tyr-171/Lys-175, respectively (Figure 4). His-173 is unique for SbFNR1, as all the other compared enzymes including SbFNR2 have tyrosine in that position. In SbFNR1, the 2′ and 3′ hydroxyl groups of the NADPH ribosyl ring were hydrogen bonded only to the sidechain of Lys-177 that was, in turn, indirectly connected to the sidechain Nδ atom of His-173 through two consecutive water molecules (Figure 4). Thus, compared to PvDFRa, the effect of Lys-177 on lowering *pK_a_* could be less significant. The hydroxyl group of Ser-133 and Nε of His-173 establish hydrogen bonds with the carbonyl oxygen of the bound naringenin, allowing for a hydride transfer between the C4 atom of the nicotinamide and the C4 atom of the substrate. Same as PvDFRa, pro-*R* hydride transfer occurs in both SbFNR1 and FNR2 and then the oxyanion of the C4 carbonyl oxygen atom is temporarily stabilized by the oxyanion hole established from the side chains of Ser-133 and His-173 (Figure 4A–C). The protonation to produce a C4 hydroxyl group could be through the imidazole sidechain of His-173. Overall, the observed lower activity of SbFNR1 and SbFNR1 against naringenin and eridyctiol (Figure 6) is likely due to its limi-ted hydrogen bond network and a longer distance for hydride transfer than that of PvDFRa in addition to their tendency to form a non-productive complex.

### 3.6. ANR Activity

To our surprise, both SbFNR1 and SbFNR2 displayed substantial ANR-activity against two anthocyanidins, cyanidin and pelargonidin (Figure 6 and Appendix A). Similar to those in ANR, the expanded substrate-binding pocket observed in SbFNR1 and SbFNR2 allows for the binding of the anthocyanidins into multiple orientations. Therefore, a hydride transfer reaction can happen on the C-ring of anthocyanidin at either the C2 atom or C3 atom (Figure 7A,B). Thus, depending on the orientation of the substrate in the initial hydride transfer reaction, four possible stereoisomers can arise. As observed in ANR, the conversion of anthocyanidins to their corresponding flavan-3-ols requires the use of two NAD(P)H molecules per anthocyanidin molecule. Thus, it is likely that all four products of the first hydride transfer reaction dissociate to replace NADP^+^ with NAD(P)H and then reassociate to the active site of SbFNR2 with a similar ΔG (Figure 7C–F). The corresponding binding energy differences to SbFNR2 given by the molecular docking were all within 1 kcal⋅mol^−1^, therefore, the production rate of certain flavan-3-ol stereoisomers could be determined by the effectiveness of the two hydride transfer reactions. The production of multiple stereoisomers could contribute to the large diversity of proanthocyanidins where catechin and epicatechin are the main components [62]. Two 2R configuration products, l-epicatechin and d-catechin, appeared from mass spec as most abundant (Appendix A) and were confirmed via standards in l-epicatechin and d-catechin. As shown in the two docked configurations of cyanidin in the electrostatic potential map in SbFNR2 (Figure 7A,B), the oxyanion hole established by the catalytic serine orients the transition state properly and thus allows the hydride transfer to the 2R confirmation.

### 3.7. Stress and Overexpression

Specific genes linked to the phenylpropanoid and flavonoid pathways can result in the accumulation of anthocyanidins and/or flavonoids (Figure 9) [63,64]. To better discern the potential co-regulation of genes involved in these pathways, previously published transcriptomic data from switchgrass and sorghum plants [65,66,67,68,69,70] were mined for expression of DFR, ANS, and ANR copies annotated in the respective genomes (Figure 9). In switchgrass, infestation with greenbugs (*Schizaphis graminum*; GB) [66], caused a strong upregulation of ANR and FNR at 10 and 15 days post infestation, suggestive of a role for anthocyanidins and/or flavan-3-ols as part of the switchgrass defense responses to GB herbivory. Similarly, for hybrid switchgrass generated from crosses between upland ‘Summer’ and lowland ‘Kanlow’ cultivars infested with GB or the yellow sugarcane aphid (*Sipha flava*; YSA) [70], PvDFRa expression was induced by GB herbivory, and several copies of ANR and FNR were strongly induced at 15 days post infestation (Figure 9). YSA infestation induced a significant overexpression of DFR, and several copies of ANR and FNR, especially at 15 days post infestation (Figure 9). Several flavonoids accumulated in hybrid switchgrass plants infested with YSA 15 days post infestation [70], coincident with enhanced gene expression of key flavonoid genes. Fall armyworm (*Spodoptera frugiperda*; FAW) infestation similarly induced the expression of several copies of DFR, ANR, and FNR, especially in the upland cultivar ‘Summer’, although some induction of expression of DFR and ANR copies were observed in ‘Kanlow’ plants infested with FAW (Figure 9). These data indicate that differences in gene expression observed for switchgrass cultivars subjected to herbivory by different insects could underlie more nuanced roles for specific flavonoids in plant defense. Conceivably, ANR activity could be key to generating flux towards defense-related flavonoids and providing some evidence for potential metabolic channeling via macromolecular clusters [71]. In sorghum, the sugarcane aphid (*Melanaphis sacchari*; SCA) strongly induced many genes relevant to flavonoid and anthocyanidin biosynthesis in the susceptible line BCK60, but not to a similar extent in the resistant line RTx2783 [69], suggesting that the extent of flavonoid pathway contributions to plant defense against SCA is variable, and driven by the genetic make-up of the plant. Evidence from overexpression (OE) of specific genes in sorghum appears to bear out the influence of plant genetics as one of the drivers for flavonoid biosynthesis. OE of MYB transcription factor SbMYB60 either upregulated or downregulated specific genes linked to these pigment pathways relative to the wild-type plants [65]. The strength of predicted MYB60 activity (based on OE levels relative to WT) appeared to influence flavonoid pathway genes, but not in a consistent manner (Figure 9). Plausibly, a lack of obvious patterns in gene co-expression could arise from differences in the amount of phenylpropanoid intermediates utilized for lignin biosynthesis versus flavonoid biosynthesis in the MYB60 OE lines (Scully et al., 2018). A more direct observation on the impact of increasing cinnamoyl-CoA intermediates was observed in sorghum plants overexpressing *CCoAOMT* (Figure 9). A likely increase in available CoA-substrates in the *CCoAOMT* OE lines impacted the upregulation of *DFR*, *ANR*, and *FNR*, especially in the stems, although one copy of *ANR* was strongly upregulated in the leaves of line 9a (Figure 9). Together these data suggest a dynamic regulation of the flavonoid pathway and indicate that DFR, ANR, and FNR appear to be crucial to plant defense.

## 4. Materials and Methods

Chemicals and Software: Analytical-grade chemicals were obtained from Sigma-Aldrich (St. Louis, MO), Thermo Fisher (Waltham, MA, USA), and Alfa-Aesar (Ward Hill, MA, USA). Additionally, cyanidin, delphinidin, and pelargonidin were obtained from Cayman Chemical (Ann Arbor, MI, USA). Screening solutions for crystallization were obtained from Hampton Research (Aliso Viejo, CA, USA). Molecular graphics images were produced using the Chimera package (University of California San Francisco, NIH P41 RR-01081). The plotted figures were generated by GraphPad Prism (GraphPad Software, Inc., San Diego, CA, USA). OriginX was used for analysis of ITC data (OriginLab Corporation, Northampton, MA, USA). Structural alignments were made using Jalview (University of Dundee).

*Expression and purification of recombinant PvDFRa, SbDFR3, SbFNR1, and FNR2*: PvDFRa complementary DNA (cDNA) representing the Panicum virgatum LOC120707535 (Pavir.5KG450100) was modified to encode an N-terminal 6×-His tag and was cloned into a pET-30a (+) vector (EMD Millipore, St. Louis, MO, USA) for overexpression. The plasmid was introduced into Escherichia coli BL21(DE3) cells (EMD Millipore, St. Louis, MO, USA). Five mL of lysogeny broth was incubated overnight with stock cells with 50 µg mL^−1^ kanamycin prior to being transferred to 2–1.5 L lysogeny broth with 50 µg mL^−1^ kanamycin at 37 °C in an orbital shaker until OD600 = 0.6. The temperature was then reduced to 25 °C and isoprophylthio-β-galactoside was added to a final concentration of 0.5 mM. Cells were induced for 12 h and harvested using centrifugation at 8000× *g* for 15 min at 4 °C. Buffer A (50 mM Tris, pH 8.0, 300 mM NaCl) was used to suspend cells prior to sonication on ice for 30 min (model 450 Sonifier, Branson Ultrasonics, Brookfield, CT, USA). The homogenized fraction was applied to centrifugation at 31,000× *g* for 1 h. The supernatant was immediately applied to a nickel-NTA column (Qiagen, Hilden, Germany) and washed with Buffer A containing 20 mM imidazole using 10 column volumes. The recombinant protein was eluted using buffer A containing 250 mM imidazole. The eluted protein was buffer exchanged to Buffer B (10 mM Tris, pH 8, 50 mM NaCl and 5% (*v*/*v*) glycerol) after concentration using an Amicon 8050 ultrafiltration cell (30-kD cutoff membrane, EMD Millipore, Burlington, MA USA), followed by overnight dialysis using a 30kD membrane bag into Buffer B. The protein was applied at a flow rate of 3 mL min^−1^ to a Mono-Q Column (GE Healthcare, Chicago, IL USA) that was pre-equilibrated with Buffer B. The protein was eluted using Buffer B containing 100 mM NaCl. The resulting fraction was concentrated and injected into a HiLoad 16/600 Superdex 200 column (Cytiva) using Buffer C (10 mM Tris pH 8.0 50 mM NaCl) at a flow rate of 1 mL min^−1^. The presence of PvDFRa in eluted fractions was confirmed using SDS-PAGE and fractions were pooled and concentrated to 10 mg mL^−1^ using BSA Assay Kit (Thermo Fisher). Purity was estimated to be greater than 99%.

SbDFR3 complementary DNA (cDNA) representing the Sorghum bicolor Sobic.004G050200 was modified to encode an N-terminal 6×-His tag and was cloned into a pET-30a (+) vector (EMD Millipore, St. Louis, MO) for overexpression. The plasmid was introduced into *Escherichia coli* BL21(DE3) cells (EMD Millipore, St. Louis, MO, USA). Five mL of lysogeny broth was incubated overnight with stock cells with 50 µg mL^−1^ kanamycin prior to being transferred to 2–1.5 L lysogeny broth with 50 µg mL^−1^ kanamycin at 37 °C in an orbital shaker until OD600 = 0.6. The temperature was then reduced to 25 °C and isoprophylthio-*β*-galactoside was added to a final concentration of 0.5 mM. Cells were induced for 12 h and harvested using centrifugation at 8000× *g* for 15 min at 4 °C. Buffer A (50 mM Tris, pH 8.0, 300 mM NaCl) was used to suspend cells prior to sonication on ice for 30 min (model 450 Sonifier, Branson Ultrasonics). The homogenized fraction was applied to centrifugation at 31,000× *g* for 1 h. The supernatant was immediately applied to a nickel-NTA column (Qiagen) and washed with Buffer A containing 20 mM imidazole using 10 column volumes. The recombinant protein was eluted using buffer A containing 250 mM imidazole. The eluted protein was buffer exchanged to Buffer B (10 mM Tris, pH 8, 50 mM NaCl, and 5% (*v*/*v*) glycerol) after concentration using an Amicon 8050 ultrafiltration cell (30-kD cutoff membrane, EMD Millipore), followed by overnight dialysis using a 30 kD membrane bag into Buffer B. The protein was applied at a flow rate of 3 mL min^−1^ to a Mono-Q Column (GE Healthcare) that was pre-equilibrated with Buffer B. The protein was eluted using Buffer B containing 100 mM NaCl. The resulting fraction was concentrated and injected into a HiLoad 16/600 Superdex 200 column (Cytiva) using Buffer C (10 mM Tris pH 8.0 50 mM NaCl) at a flow rate of 1 mL min^−1^. The presence of PvDFRa in eluted fractions was confirmed using SDS-PAGE and fractions were pooled and concentrated to 10 mg mL^−1^ using BSA Assay Kit (Thermo Fisher). Purity was estimated to be greater than 95%.

SbFNR cDNA sequence corresponding to *Sobic.006G226800* (*Sb06g029630*) and *Sobic.006G226700.1* were cloned into vector pET-30a (+) and introduced into *Escherichia coli* BL21(DE3) cells via transformation. Three-liter LB medium complemented 50 μg mL^−1^ kanamycin was inoculated with 20 mL from an overnight culture. The cells were grown at 37 ℃ until the culture reached OD_600_ = 0.4, and the temperature was reduced to 16 ℃ prior to the addition of IPTG to a final concentration of 0.5 mM. After being induced for 16 h, the cells were harvested by centrifugation at 8000× *g* for 10 min at 4 ℃. The cells were resuspended by buffer A with 20 mM imidazole and sonicated to release proteins. The crude lysate was centrifuged (15 min, 13,000× *g*), and the supernatant containing the soluble proteins was loaded on a nickel-NTA column as described for PvDFRa. The protein was concentrated with an Amicon 8050 ultrafiltration cell (30-kD cutoff membrane). The buffer was exchanged against 5 mM potassium phosphate buffer and loaded onto a hydroxyapatite column. The fraction containing SbFNR was eluted by a linear gradient of 5% to 10% (w/v) potassium phosphate and concentrated to 1 mL. Then it was loaded onto Superdex™ 200 Increase 10/300 GL for further purification. The purified SbFNR was buffer exchanged against 20 mM Tris, pH 7.5, and 50 mM NaCl (Buffer D) for crystallization.

Site-directed mutagenesis was conducted using primers generated with the QuikChange Primer Design (Agilent Santa Clara, CA USA).

### Crystallization and Structure Determination

*PvDFRa*: For the crystallization of PvDFRa, a solution of pure PvDFRa (10 mg mL^−1^) in Buffer C was prepared. Crystallization trials were performed using the hanging-drop vapor-diffusion method at a temperature of 277 K. PvDFR crystals were obtained by mixing the above protein solution with an equal volume of reservoir solution containing 20% (*w*/*v*) PEG 3350 and 0.1 M Tris (pH 8.5). Diffraction-quality crystals usually appeared after 2 days and larger rod-shaped crystals with dimensions of approximately 2 mm × 0.5 mm × 0.5 mm were obtained after 7 days. To obtain the complex crystal of DHQ with DFR, crystals grown as described above were soaked in 1mM DHQ mother liquor with a cryoprotectant (25% (*v*/*v*) glycerol) for 30 min. Crystals of recombinant PvDFR were grown using the hanging-drop, vapor-diffusion method. Purified PvDFR was concentrated to 7 mg mL^−1^ in 10 mM Tris pH 8, 50 mM NaCl and mixed in a 1:1 volumetric ratio with the solution from the reservoir, 0.1 M Tris-HCl pH 8.5, 25% PEG 3350. Complex crystals were obtained by soaking apo-crystals in a 1 mM DHQ solution of the mother liquor for 30 min. followed by transfer to cryoprotectant. Diffraction data for NADP^+^ and DHQ complex crystals were collected at Advanced Light Source in Beamline 5.0.1 and 8.2.2 at 100 K.

*SbFNR*: Prior to crystallization, the SbFNRs were concentrated to 20 mg mL^−1^ by using an Amicon 8050 ultrafiltration cell with a 10-kDa cutoff membrane. A solution of 1 mM NADP^+^ was added to the SbFNR1 protein solution and 1mM naringenin was added to the SbFNR2 protein solution. A commercial crystallization kit, Crystal Screen HT (Hampton), was used for crystal screening through the sitting-drop, vapor-diffusion method by Crystal Phoenix (Art Robbins Instruments, Sunnyvale, CA, USA). The initial crystal of SbFNR1 NADP^+^ or naringenin complex appeared in the solution of D3 (0.1 M HEPES, pH 7.5, 2 M Ammonium Sulfate and 2% PEG 400) at 4 °C, while the SbFNR2 naringenin complex crystals appeared in G8 (0.1 M sodium chloride, 0.1 M HEPES pH 7.5, 1.6 M ammonium sulfate). Then the larger crystals were reproduced by the hanging-drop vapor-diffusion method. Large crystals appeared in 2–14 days. The ternary complex crystals of SbFNR1 with NADP(H) and naringenin were obtained by soaking the naringenin binary complex crystal in the mother liquor containing 10 mM NADPH or NADP^+^, while the DHQ complex for SbFNR1 was obtained by soaking the NADP^+^ complex in mother liquor containing 10 mM DHQ for 10 min. The complex SbFNR2 crystal with naringenin and NADPH was obtained by soaking the naringenin complex in a solution of 10 mM NADPH for 10 min. Data were collected at the Advanced Light Source beamline 5.0.1. The software package HKL2000 was used for diffraction data processing [72].

Initial phasing of PvDFRa diffraction data was performed by molecular replacement with the PHENIX Phaser [73] using the coordinates of DFR from *Vitis vinifera* (PDBI: 2C29) as the search model. The phasing of SbFNR was performed by molecular replacement with PvDFRa coordinates. PHENIX and Coot [74] were used for refinement and model building, respectively. The diffraction data statistics are listed in Table 1. Crystallographic data and coordinates were deposited in the Protein Data Bank.

Kinetic Assays of PvDFRa, SbDFR3, mutant PvDFRa, and SbFNRs: DFR was purified to the same purity as used in crystallization. Enzymatic assay followed the methodology previously reported [47] with small changes. A solution of 0.1 M potassium phosphate at pH 7.0 was utilized, and substrate concentration was varied from 5 µM to 800 µM. The reaction was initiated with 0.05 mg enzyme in a total reaction volume of 200 µL. To assay the conversion of DHK, DHQ, and DHM, the concentration of NADPH was held constant at 2 mM. To assay the dependency of the cofactor, the concentration of DHQ was held constant at 1 mM and the concentrations of NADPH and NADH were varied from 5 to 500 µM. The reaction proceeded for 15 min. at 25 °C and was terminated by the addition of 600 µL 95:5 (*v*/*v*) *n*-butanol:HCl. The products were then heated to 95 °C for 1 h, which converted the unstable leucoanthocyanidins to the more stable corresponding anthocyanidins. The reactions were centrifuged for 15 min. at 14,000× *g* and the supernatant was used to determine product concentrations with a GENESYS™ 10S UV-Vis Spectrophotometer (Thermo Scientific, Waltham, MA, USA). The absorbances of cyanidin delphinidin and pelargonidin were measured at 550, 550, and 528 nm, respectively. The concentrations were determined based on calibrations with reference compounds (Appendix A). Flavanones were assayed utilizing the same general method, but the heating step was reduced to 3 min due to the lower product stability. Data analysis was conducted utilizing GraphPad Prism 8.0.2 using linear regression to model the appropriate model. Error in each data point is shown utilizing the standard deviation.

To observe the inhibition of PvDFRa activity by quercetin, the concentration of DHM was held constant at 1 mM and two concentrations of quercetin at 75 and 150 μM. The referenced *V_max_* and *K_m_* values were used to make a nonlinear regression. Quercetin was assayed for its potential for reduction at the carbonyl oxygen by reacting 500 µM quercetin in 0.1 M phosphate buffer with 2 mM NADPH for 30 min. Five µL of the reaction was injected into a reversed-phase HPLC column XBridge Peptide BEH C18 (Waters, Milford, MA, USA). Acetonitrile:water was used as the mobile phase, using a gradient in the concentration of acetonitrile from 5% (*v*/*v*) to 100%.

The FNR activity assay was established based on a previous SbFNR study [32]. The reaction mixtures contained 0.1 M potassium phosphate buffer (pH 7.0), 2 mM NADPH, and varying concentrations of substrate (naringenin or eriodictyol) with a final volume of 200 μL. The reaction was initiated by the addition of 30 ng SbFNR and continued for 16 h at 30 ℃. The reaction was then terminated by adding 200 μL ethyl acetate. The top layer was removed and dried down, and the residual was dissolved in 2 N HCl in 50% (*v*/*v*) methanol. The acid treatment was conducted to convert the direct reaction products, flavan-3,4-diols, and flavan-4-ols, to their corresponding anthocyanidins or 3-deoxyanthocyanidins, as evidenced by the formation of a pink color. The sample was incubated at 80℃ for 3 min. to assay calorimetrically, dried, and dissolved in 200 μL of 0.1% (*v*/*v*) formic acid in water for LC-MS analysis [75]. Data analysis was conducted utilizing GraphPad Prism 8.0.2 using linear regression to model the appropriate model. Error in each data point is shown utilizing the standard deviation.

LC-MS was used to analyze the various flavonoids using a Waters Xevo TQ-MS mass spectrometer interfaced with a Waters Acquity UPLC. An Ace Excel 1.7 SuperC18 (P/N EXL-1711-1003U, 100 mm × 3.0 mm i.d.) reverse phase HPLC column was used. The chromatographic conditions used were based on [75]. The mobile phase used was a binary gradient of water with 0.1% (*v*/*v*) formic acid (Solvent A) and methanol with 0.1% formic acid (Solvent B). The solvent flow rate was 0.2 mL min^-1^. and the initial solvent composition was 20% B/80% A and was constant for the first minute following sample injection. Gradient elution was accomplished by increasing the ratio of solvent B to solvent A from 20:80 to 65:35 over 10 min. using a linear gradient, and then increased to 100:0 over the next 4 min. The composition was maintained at 100% B for 5 min., returned to 80%A/20%B over the next 3 min., and held at that ratio for the next 3 min. The mass spectrometer was operated in positive ion mode with a capillary voltage of 3.2 V and a cone voltage of 17 V (unless specified differently in specific MS methods). The source temperature was 350 °C and API gas flow was 650 L h^-1^.

Appropriate daughter ions to be used for multiple reaction monitoring (MRM) of (-)-epicatechin were identified by infusing standard and recording spectra. The optimum cone voltage and collision energy of epicatechin were also determined in these infusion experiments. MRM transitions and appropriate collision energies for the flavonoid derivatives were found on the MassBank of North America website https://mona.fiehnlab.ucdavis.edu/ (accessed on 13 December 2022). Single isotope reaction (SIR) methods were used for the analysis of the hypothesized product ions to monitor. The resulting peaks were integrated using TargetLynx data analysis software 4.2 (Waters Corp, Milford, MA, USA). Data analysis was conducted utilizing GraphPad Prism 8.0.2 using linear regression to model the appropriate model. Error in each data point is shown utilizing the standard deviation.

Isothermal Titration Calorimetry: Purified PvDFR was dialyzed to a buffer containing 50 mM Bis-Tris, 50 mM Tricine (pH 6.5), 200 mM NaCl and 2.5% (*v*/*v*) glycerol. Substrates were dissolved in the same buffer used for enzyme dialysis. Analysis was conducted utilizing a Microcal ITC 200 (Malvern). Concentrations for substrate varied depending on binding affinity at a temperature of 298 K. For NADP^+^, the concentration was 1 mM; all other substrates were used at concentrations of 1.25 mM. A total of 22 injections were made, each with a volume of 2 uL. Data analysis was conducted utilizing Origin Software 7 and modeled to the appropriate nonlinear regression.

Molecular Docking: Initial screening of the enzyme was conducted by establishing a grid of 50 Å × 50 Å × 50 Å in Autodock Tools. The screening was conducted using a grid centered at 57.408, 29.726, and 13.193 with 25 as the search exhaustiveness. The resulting positions identified the binding pocked previously reported from VvDFR as the highest affinity location. Therefore, a 20 Å × 20 Å × 20 Å pocket was established at points 57.408, 29.726, and 13.193 and the highest affinity binding position was located dependent on the position of C4 of the nicotinamide being a reasonable distance from the carbonyl carbon of the substrate.

Phylogenetic Tree: A phylogenetic tree was made by initially utilizing the Basic Local Alignment Search Tool (BLAST; Altschul et al., 1997)) utilizing non-redundant protein sequences against PvDFRa. Alignment was initially conducted in Molecular Evolutionary Genetics Analysis 11 (MEGA 11; [76]) utilizing ClustalW [77]. A phylogenetic tree was then created utilizing the Jones–Taylor–Thornton substitution matrix [52]. The ancestral sequence was defined via MEGA11 and set as the out–group.

Accession Numbers

Sequence data from this publication can be found in the EMBL/GenBank data libraries under accession numbers Pavir.5KG450100 (PvDFRa), Pavir.5NG271200 (PvDFRb), EES11482.1 (SbFNR1) and XP_021318468.1 (SbFNR2). The structure discussed in this manuscript can be found at www.rcsb.org deposited under the corresponding PDB IDs: 8FEM, 8FEN, 8FEU, 8FEV, 8FET, 8FEW, 8FIO, 8FIP.

## 5. Conclusions

Flavonoids are phenolic compounds that protect plants against pathogens, UV radiation and help attract pollinators. Several of these compounds have generated interest as nutraceuticals and anti-cancer drugs. The 3-deoxyanthocyanidins (3-DOA) of sorghum are of special interest because they are not present in other cereals and their biosynthesis is not yet fully elucidated. DFR, ANR, and FNR are the enzymes in the central step in flavonoid biosynthesis that is also closely connected to the monolignol biosynthetic pathway. The ability to reroute metabolic flux through the flavonoid pathway in sorghum and switchgrass via genetic approaches first requires a detailed understanding of enzyme kinetics, the catalytic mechanism, and substrate specificity. This information aids the design of enzymes with altered substrate specificity or greater catalytic efficiencies. This study established the enzyme kinetics of DFR, FNR, and ANR for eight substrates in the flavonoid pathway. DFR appears to have broad substrate specificity for dihydroflavonols, flavanones, and anthocyanidins that overlap with those of ANR and FNR. Enhanced expression of *SbDFR* is expected to increase the pool of flavan-4-ols and eventually 3-DOA, to enhance the defense against pathogens and to increase the market value of sorghum seed. The combined information from this study will aid the design of enzymes with altered substrate specificity or greater catalytic efficiencies as a way to redirect metabolic flux to specific compounds of value.

## Figures and Tables

**Figure 1 ijms-24-13901-f001:**
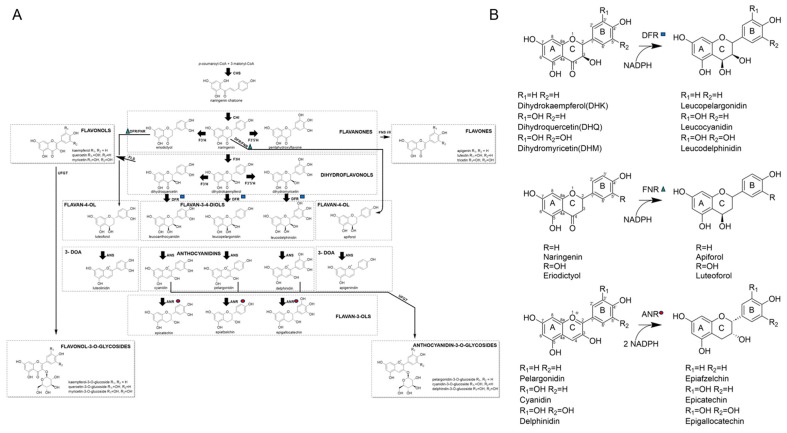
(**A**) Flavonoid pathway. Starting with *p*-coumaroyl CoA from the monolignol pathway to yield the typical 15-carbon skeletal structure. CHS, Chalcone synthase; CHI, chalcone isomerase; F3′H, flavonoid 3′-hydroxylase; F3′5′H, flavonoid 3′5′-hydroxylase; F3H, flavanone 3-hydroxylase; DFR, dihydroflavonol 4-reductase; FNR, flavanone 4-reductase; ANS, anthocyanidin synthase; FNSI/II, flavone synthase I/II; FLS, flavonol synthase; ANR, anthocyanidin reductase; UFGT, UDP-glucose flavonoid 3-*O*-glucosyltransferase. (**B**) The primary catalytic activity of DFR/FNR/ANR in the flavonoid pathways. dihydroflavonols to flavon-3,4-diols, flavanones to flavan-4-ols, and anthocyanidins to flavan-3-ols. B-ring hydroxylation differs between the substrates for each enzymatic reaction. The blue square, green triangle, and red circle denote reactions of DFR, FNR, and ANR, respectively.

**Figure 2 ijms-24-13901-f002:**
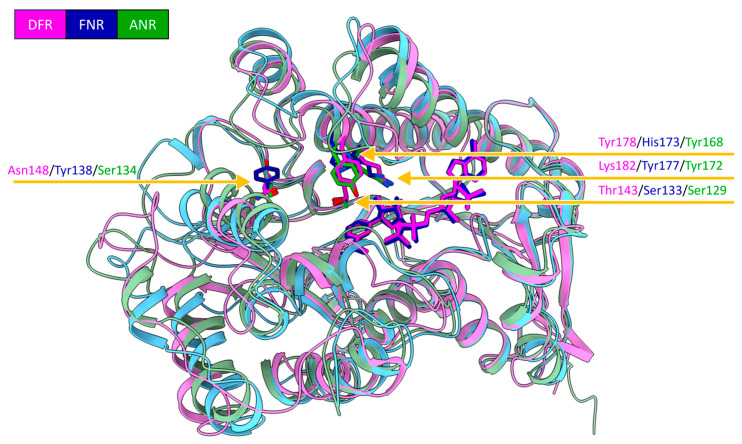
Superimposed tertiary structures of PvDFRa (pink), SbFNR1 (blue), and SbANR (green). Catalytic triad residues, as well as bound NADP^+^ are represented as sticks. The catalytic triads are shown with residue numbers and arrows. Additionally, Asn-148/Tyr-138/Ser-131 are important for substrate specificity. RMSD among those structures is shown in Appendix A. Molecular graphics images were produced using the ChimeraX package (UCSF).

**Figure 3 ijms-24-13901-f003:**
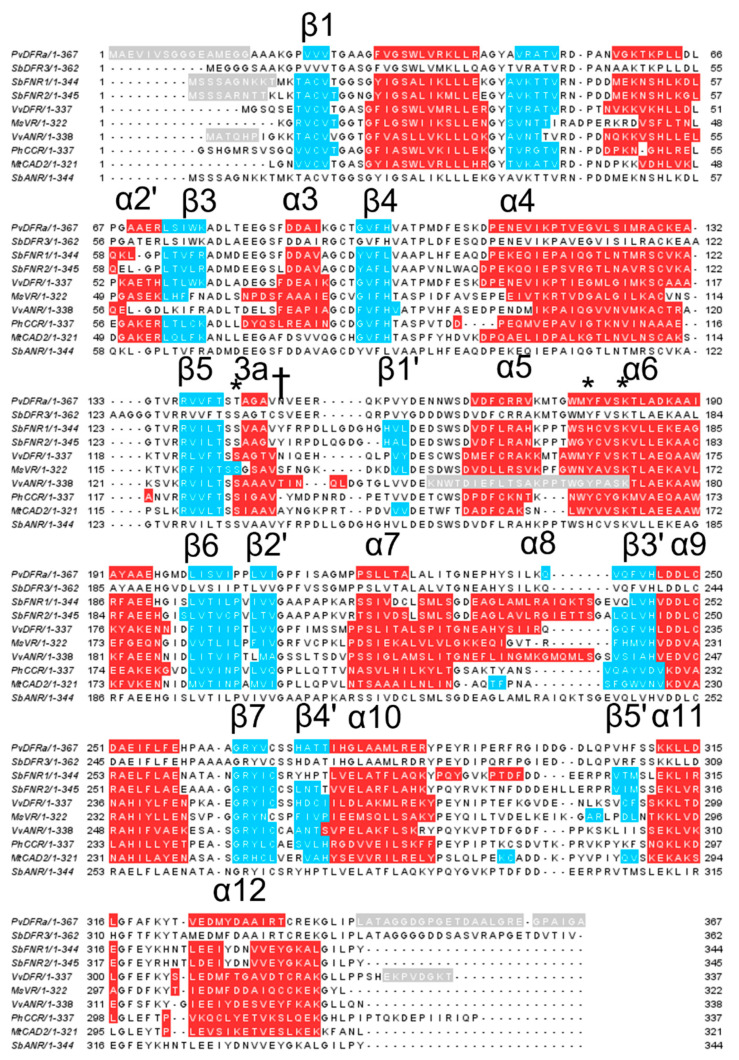
Multiple alignments with the amino acid sequences of *Panicum virgatum* DFR (PvDFRa) and close structural homologs. Included in the alignment are *Sorghum bicolor* DFR (SbDFR3), *Sorghum bicolor* FNRs (SbFNR1 and SbFNR2), *Vitis vinifera* DFR (VvDFR), *Medicago sativa* vestitone reductase (MsVR), *Vitis vinifera* anthocyanidin reductase (VvANR), *Petunia x hybrida* cinnamoyl-CoA reductase (PhCCR) and *Medicago truncatula* cinnamyl alcohol dehydrogenase (MtCAD2). Residues highlighted in blue are in β-strands, while those in red are in α-helices. A short 3_10_ helix is labeled 3a. Catalytic triad residues are indicated with an asterisk (*) and the residue facing the B-ring is indicated with a dagger (†).

**Figure 4 ijms-24-13901-f004:**
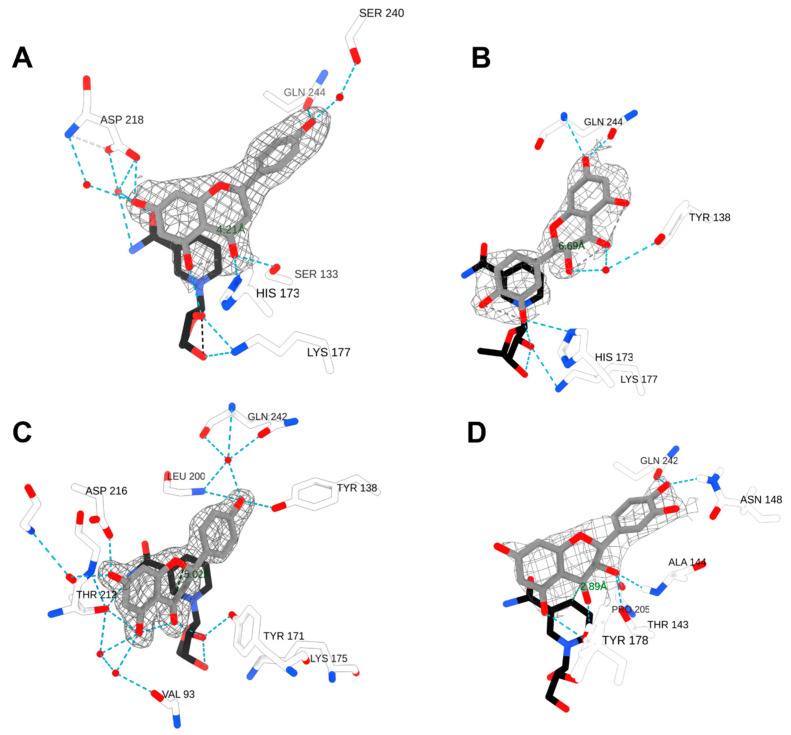
The active sites of SbFNR1, FNR2, and PvDFRa with bound ligands. Ligands are shown in grey, NADP is shown in black, interacting residues are shown in white, hydrogen bonds are shown via dashed lines, oxygen is shown in red, and nitrogen is shown in blue, and the density covering the ligand is displayed in grey mesh using the *Fo-Fc* map. (**A**) the complex crystal structure of SbFNR1 with naringenin and NADP^+^ (PDBID: 8FEU); (**B**) the complex crystal structure of SbFNR1 with DHQ and NADP^+^ (PDBID: 8FEV); (**C**) the complex crystal structure of SbFNR2 with naringenin and NADP^+^ (PDBID: 8FIO). (**D**) the complex crystal structure of PvDFRa with DHQ and NADP^+^ (PDBID: 8FEN). Electron density is shown at 1.1 RMSD contour level. Molecular graphics images were produced using the ChimeraX package (UCSF).

**Figure 5 ijms-24-13901-f005:**
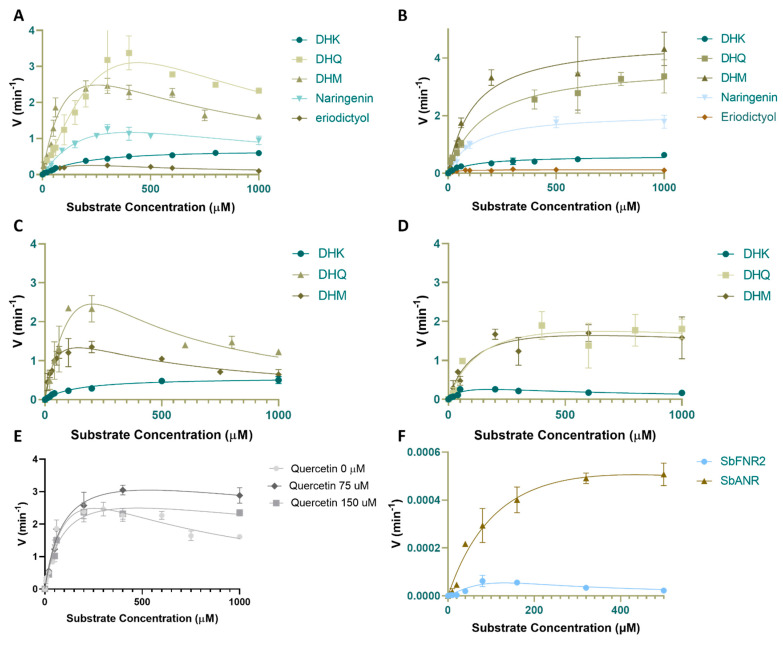
Enzyme kinetic assays for DFR, DFR, and ANR: (**A**) Michaelis-Menten curves for wild-type PvDFRa for various substrates. The curves were constructed using initial rate measurements. For each reaction, the concentration of NADPH was held constant at 2 mM and the concentrations of substrates were varied. All experiments were conducted in triplicate (*n* = 3). PvDFRa was most active with DHQ and minimally active with eriodictyol. (**B**) Michaelis-Menten curves for SbDFR3 under the same conditions. (**C**) Michaelis-Menten curves for PvDFRa T143S mutant under the same conditions. (**D**) Michaelis-Menten curves for PvDFR Q242A mutant under the same conditions. (**E**) Michaelis-Menten curves for cyanidin conversion to all four isomers of catechin in SbANR and SbFNR2. (**F**) Steady-state initial rates plotted against DHM concentration with varying concentrations of quercetin acting as an inhibitor. The data were processed using GraphPad Prism 8.0.2 (San Diego, CA, USA).

**Figure 6 ijms-24-13901-f006:**
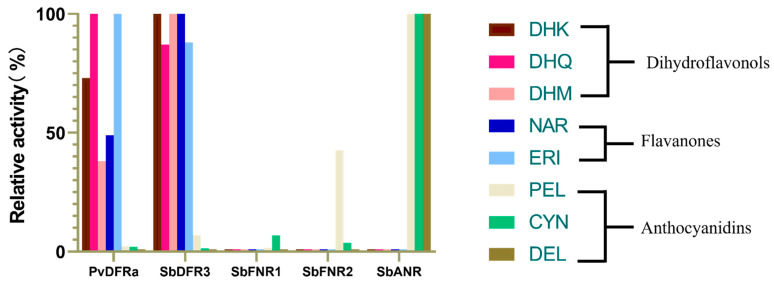
Relative activities of the five oxidoreductases. The activity was compared based on the relative activity of the reductase of the highest activity. Dihydroflavonols are shown in red hues assigning the activity of SbDFR or PvDFRa as 100%, flavanones are shown in blue hues assigning SbDFR or PvDFRa as 100%, and anthocyanidins assigning SbANR as 100% are shown in green hues.

**Figure 7 ijms-24-13901-f007:**
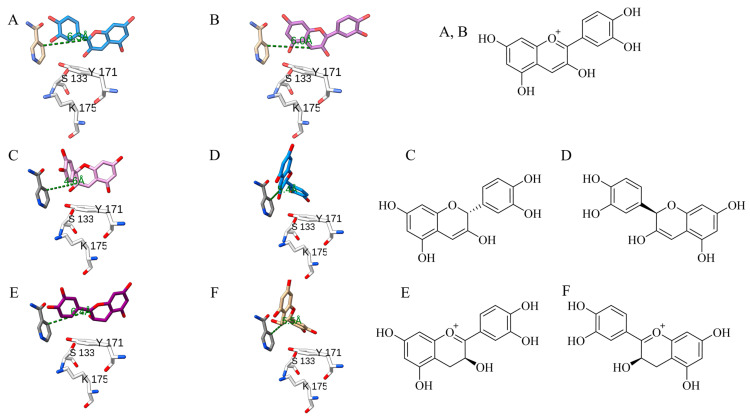
Docked positions of substrates at the active site: Orientation of cyanidin at the active site of SbFNR2 for the first hydride transfer from NADPH to (**A**) C2 atom of cyanidin, (**B**) C4 atom of cyanidin. (**C**–**F**) the resulting stereoisomers of the first hydride transfer reaction, which are associated with the active site for the secondary hydride transfer. (**A**–**F**) **in the right-side panel** the compounds in the right-side panel are drawn with ChemDraw v.22.2.0. (**A**,**B**) cyanidin. IUPAC nomenclature of the corresponding intermediates and products are (**C**) (*R*)-2-(3,4-dihydroxyphenyl)-2H-chromene-3,5,7-triol; (**D**) (*S*)-2-(3,4-dihydroxyphenyl)-2H-chromene-3,5,7-triol; (**E**) (*S*)-2-(3,4-dihydroxyphenyl)-3,5,7-trihydroxy-3,4-dihydrochromenylium; (**F**) (*R*)-2-(3,4-dihydroxyphenyl)-3,5,7-trihydroxy-3,4-dihydrochromenylium. The figure was generated by ChimeraX (UCSF).

**Figure 8 ijms-24-13901-f008:**
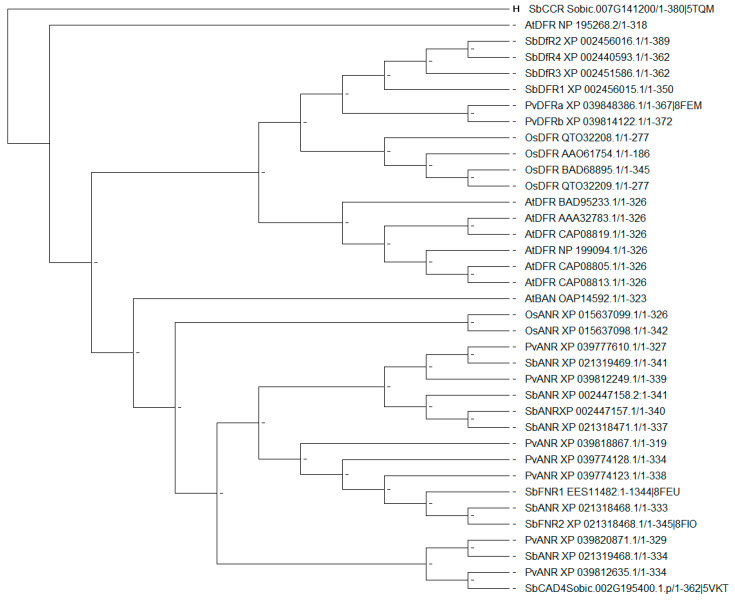
Phylogenetic Tree for DFR, FNR, and ANR. A BLASTP search with non-redundant protein sequences in *P. virgatum*, *S. bicolor*, *O. sativa*, and *A. thaliana* and a phylogenetic tree with Molecular Evolutionary Genetics Analysis (MEGA11) using the Jones–Taylor–Thornton (JTT) substitution model [52]. Enzymes are annotated with gene names and PDBID if applicable. H indicates the ancestrally inferred protein from the series of enzymes.

**Figure 9 ijms-24-13901-f009:**
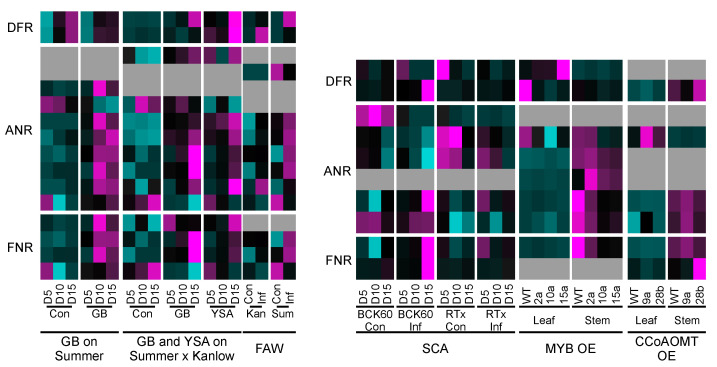
DFR, ANS, and, ANR expression in switchgrass and sorghum. A heatmap showing the relative expression patterns of flavonoid biosynthesis pathway genes in switchgrass (**left**) and sorghum (**right**). The switchgrass datasets include cv ‘Summer’ plants infested with greenbugs (GB; *Schizaphis graminum*) [66], hybrid switchgrass plants (cv ‘Summer’ × cv ‘Kanlow’) infested with GB or yellow sugarcane aphids (YSA; *Sipha flava*) [70], and ‘Kanlow’ (Kan) or ‘Summer’ (Sum) plants infested with fall army worm (FAW; *Spodoptera frugiperda*) [68]. Sorghum datasets include sugarcane aphids (SCA) infestation of plants from a susceptible sorghum line BCK60 and a resistant line RTx2783 [69], expression of genes in leaves (leaf) and stems (stem) in wild-type (WT) and three sorghum *MYB60* over-expression lines [67], and in WT and two *CCoAOMT* over-expression lines [67]. Gene expression was standardized to z-scores separately for each experiment. Magenta = high expression, black = intermediate expression, cyan = low/no expression, grey = no detectable expression within an experiment. Gene abbreviations are the same as in Figure 1. In all panels when present, con = controls; D5, D10, and D15 refer to sampling dates within each experiment. Specific details can be found in the respective cited references.

**Table 1 ijms-24-13901-t001:** X-ray diffraction data and refinement statistics for PvDFRa, SbFNR1 and SbFNR2.

	PvDFRa (NADP^+^ Complex)	PvDFRa (NADP^+^/DHQ Complex)	SbFNR1 (NADP^+^ Complex)	SbFNR1 (Naringenin Complex)	SbFNR1 (NADP(H)/Naringenin Complex)	SbFNR1 (NADP^+^/DHQ Complex)	SbFNR2 (NADP(H)/Naringenin Complex)	SbFNR2 (NADP^+^ Complex)
PDB ID	8FEM	8FEN	8FET	8FEW	8FEU	8FEV	8FIO	8FIP
Data collection	ALS 5.0.1	ALS 8.2.2	ALS 5.0.1	ALS 5.0.3	ALS 5.0.1	ALS 8.2.2	ALS 5.0.1	ALS 5.0.1
Space group	P 42 21 2	P 42 21 2	I 41 2 2	I 41 2 2	I 41 2 2	I 41 2 2	P 21 21 21	P 2 21 21
Cell dimensions								
a, b, c (Å)	119.177 119.177 56.906	119.177 119.177 56.907	163.203 163.203 206.566	165.069 165.069 208.599	165.180 165.180 207.655	165.012 165.012 209.116	59.089 112.787 122.808	86.628 90.219 103.76
Resolution (Å)	42.14–2.34 (2.42–2.34)	33.86–2.55 (2.64–2.55)	47.14–2.20 (2.28–2.20)	46.68–2.02 (2.09–2.02)	47.44–2.12 (2.19–2.12)	47.71–2.21 (2.28–2.21)	42.58–1.97 (2.04–1.97)	66.5–1.7 (1.76–1.7)
*R*_sym_ or *R*_merge_	0.1825 (1.547)	0.2406 (0.9742)	0.105 (1.756)	0.02699 (0.754)	0.0334 (0.7037)	0.05134 (0.8971)	0.0888 (0.8098)	0.06585 (0.4378)
*I/*σ*I*	14.36 (1.26)	9.14 (1.75)	13.96 (1.77)	20.32 (0.96)	17.62 (1.16)	12.10 (0.90)	8.25 (0.83)	19.49 (2.51)
Completeness (%)	98.63 (96.84)	99.31 (98.60)	99.34 (98.61)	99.88 (99.92)	99.97 (99.96)	99.95 (99.86)	93.84 (61.70)	99.98 (99.99)
Redundancy	13.6 (9.2)	6.3 (4.8)	13.1 (12.7)	2.0 (2.0)	2.0 (2.0)	2.0 (2.0)	1.9 (1.6)	6.6 (6.7)
Refinement								
Resolution (Å)	2.34	2.55	2.20	2.02	2.12	2.21	1.97	1.70
No. reflections	241,446	87,359	70,362	93,749	80,982	71,968	55,196	596,143
Rwork Rfree	0.2059 0.2353	0.2375 0.2491	0.2067 0.2318	0.2120 0.2279	0.1991 0.2270	0.2018 0.2371	0.1928 0.2277	0.1726 0.1974
No. atoms	2688	2629	5581	5606	5727	5710	5904	6118
Protein	2523	2515	5170	5170	5170	5170	5144	5172
Ligand/ion	73	70	146	89	230	239	210	146
Water	97	44	315	371	401	375	624	850
B-factors	49.33	50.32	53.92	54.26	48.74	49.87	28.16	24.20
Protein	49.46	50.39	53.81	54.02	48.33	49.23	27.63	22.76
Ligand/ion	44.56	52.58	54.80	60.33	54.44	62.22	21.34	16.48
Water	48.24	42.68	55.34	56.55	51.73	53.21	33.98	33.83
r.m.s.d.								
Bond lengths (Å)	0.005	0.005	0.006	0.008	0.008	0.007	0.005	0.008
Bond angles (°)	0.65	0.59	0.80	0.94	0.85	0.93	0.72	0.90

r.m.s.d., Root-mean-square deviation.

**Table 2 ijms-24-13901-t002:** Thermodynamic parameters of interaction between PvDFRa/SbFNR1 and various ligands measured by isothermal titration calorimetry.

ENZYME	SUBSTRATE	K_D_ (µM)	∆H (kcal mol^−1^)	∆S (Cal mol^−1^ K^−1^)
PvDFRa	NADP^+^	0.18 ± 0.06	−9.41 ± 0.15	−0.226
	DHQ	16.87 ± 4.35	−10.77 ± 0.18	−13.7
	DHM	21.23 ± 3.26	−9.10 ± 2.45	−9.14
	DHK	26.42 ± 2.26	−6.19 ± 5.54	0.854
	Eriodictyol	64.94 ± 11.84	−6.23 ± 0.28	−1.74
	Naringenin	176.9 ± 25.3	−4.82 ± 0.39	0.986
SbFNR1	NADP^+^	13.91 ± 2.72	−1.12 ± 0.72	−15.4

**Table 3 ijms-24-13901-t003:** Enzyme kinetic parameters for wild-type and mutant forms of PvDFRa, in the presence of DHQ, DHM, DHK, eriodictyol, and naringenin.

	PvDFRa WT			PvDFRa T143S			PvDFRa Q242A			SbDFR3		
	*K*_m_ (µM)	*k*_cat_ (min^−1^)	*k*_cat*_*/K_m_* (µM^−1^ min^−1^)	*K*_m_ (µM)	*k*_cat_ (min^−1^)	*k*_cat*/_*K*_m_ (µM^−1^ min^−1^)	*K*_m_ (µM)	*k*_cat_ (min^−1^)	*k*_cat*_/*K*_m_ (µM^−1^ min^−1^)	*K*_m_ (µM)	*k*_cat_ (min^−1^)	*k*_cat*_*/K*_m_ (µM^−1^ min^−1^)
DHQ	128.6 ± 14.3	3.357 ± 0.4065	0.02610	19.04 ± 11.21	1.759 ± 0.2979	0.09230	156.8 ± 121.41	2.571 ± 0.481	0.01640	166.4 ± 28.8	3.795 ± 0.172	0.02281
DHM	150.0 ± 56.9	2.261 ± 1.382	0.01507	7.409 ± 0.2465	1.057 ± 0.1153	0.1426	119.2 ± 87.86	2.273 ± 0.134	0.01907	118.5 ± 0.2885	4.640 ± 0.289	0.03916
DHK	191.2 ± 31.4	0.7253 ± 0.06364	0.003790	137.7 ± 36.00	0.5737 ± 0.062	0.004160	74.59 ± 62.70	0.4701 ± 0.2679	0.006302	117.6 ± 22.61	0.608 ± 0.511	0.00517
Eriodictyol	246.5 ± 224.1	0.8481 ± 0.559	0.003441							40.73 ± 31.89	0.1236 ± 0.011	0.00303
Naringenin	407.6 ± 223.6	3.549 ± 1.444	0.008707							118.2 ± 14.92	2.084 ± 0.087	0.01763

## Data Availability

The data that support the findings of this study are available from the corresponding author upon reasonable request.

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
