# Peer review of "Structural Similarities and Overlapping Activities among Dihydroflavonol 4-Reductase, Flavanone 4-Reductase, and Anthocyanidin Reductase Offer Metabolic Flexibility in the Flavonoid Pathway"

_ijms, 2023, doi:10.3390/ijms241813901_

Round 1

Reviewer 1 Report

Flavonoids are potent antioxidants that play a role in defense against pathogens, UV-radiation and reactive oxygen species. Sorghum bicolor has high concentrations of 3-deoxyanothocyanidins in vivo supporting the observed high activity of SbDFR for flavonols. Mining of expression data indicated a significant induction of those three reductase genes in both switchgrass and sorghum in response to biotic stress.

Compared with existing reports, the present topic is very interesting, the method used in the research is novel and reliable, and the results are supported by the experimental evidence supplied, so the conclusion was solid. This study has significantly contributed to the research on molecular sciences, especially to the research on flavonoids. This manuscript can be accepted after minor revision. The reviewer would like to review the revision again. Apart from grammar editing, several suggestions were given on this manuscript.

1, Natural products also have a large number of various flavonoids, such as “Dragon's Blood”, Fordiae Cauliflorae Radix, x and so on. So related natural products need to be mentioned in the introduction section and their research progress needs to be referred: doi.org/10.1016/j.jep.2010.11.008; doi.org/10.1186/1752-153X-6-116; doi.org/10.3390/molecules181215134; doi.org/10.1186/1752-153X-7-126.

2, The resolution of the figures should be improved, and the format of the reference list needs to be checked again.

Minor editing of English language required

Author Response

Reviewer 1

1, Natural products also have a large number of various flavonoids, such as “Dragon's Blood”, Fordiae Cauliflorae Radix, x and so on. So related natural products need to be mentioned in the introduction section and their research progress needs to be referred: doi.org/10.1016/j.jep.2010.11.008; doi.org/10.1186/1752-153X-6-116; doi.org/10.3390/molecules181215134; doi.org/10.1186/1752-153X-7-126.

In the revised version, the reviewer’s suggestion has been inserted in the introduction addressing natural products with regards to their flavonoid levels:

Additionally, traditional medicines, including “Dragon’s Blood” (Daemonorops draco), and the Asian shrubs Fordia cauliflora and Millettia pulchra, contain high levels of flavonoids, which could contribute to their health benefits [12–14].

The resolution of the figures should be improved, and the format of the reference list needs to be checked again.

In the revised version, resolution of figures was improved following the suggestion. Apologies for the issues with citations, upon conversion to IJMS, format connection between the reference manager and the document was lost. The issue has been resolved.

Reviewer 2 Report

1. This study compares three reductases involved in flavonoid pathways in an array of properties including their protein structures and activities. Then it indicates they play roles in plant response to biotic stress. The data seem valuable to add some information in this field. Still, the problem is the authors did not describe and discuss the aim and how to apply the data in further directions, i.e., why compare them and what is the meaning of the findings? Also, the manuscript title should be more constructive and informative.

2. Figure 6, Tables 2 and 5: Significant differences between means should be added.

3. A section of statistical analysis should be added to M&M

4. The authors should improve the grammar throughout the manuscript. Many sentences are challenging to understand.

5. L24: It should be careful to use the term "significant" because it needs to follow the results analyzed by ANOVA.

6. L23-30, L543-562: Very difficult to digest and maybe need a comprehensive graph to present the mechanism and how they are involved in the flavonoid pathway and then contribute to biotic stress resistance.

7. All descriptions and conclusions according to pure bioinformatics should be more conservative.

8. M&M: This should be improved because the current version is difficult to follow, maybe a flowchart will be helpful to understand the entire experimental design.

Author Response

Reviewer 2

  1. This study compares three reductases involved in flavonoid pathways in an array of properties including their protein structures and activities. Then it indicates they play roles in plant response to biotic stress. The data seem valuable to add some information in this field. Still, the problem is the authors did not describe and discuss the aim and how to apply the data in further directions, i.e., why compare them and what is the meaning of the findings? Also, the manuscript title should be more constructive and informative.

Authors appreciate these constructive suggestions. We have inserted statements about the aim, findings and future directions as follows.

Line 32-34: Key signature sequences for proper DFR/ANR classification are proposed and could form the basis for future metabolic engineering of flavonoid metabolism. 

Line 159-161: Our analysis reveals substrate preference, kinetic profiles and participating residues, inhibition patterns and overlapping activity among DFR, FNR and ANR, which offers prospects for rerouting metabolic flux towards compounds of specific interest.

Line 1062-1064: The combined information ultimately aids the design of enzymes with altered substrate specificity or greater catalytic efficiencies as a way to redirect metabolic flux to specific compounds of value.

Following the suggestion, the title has been updated to convey more information about the article.

  1. Figure 6, Tables 2 and 5: Significant differences between means should be added.

The authors followed a convention of the field for both Table 2 and Figure 5 with corresponding standard deviation (n =3). Fig. 6 is a bar graph comparing the average activity of different enzymes intended to help the readers to grab a whole trend of five enzymes and eight substrates. The corresponding standard deviations are reported in each enzyme kinetics figures (Fig. XYZ).

  1. A section of statistical analysis should be added to M&M

 Additional information has been added to the appropriate sections where data analysis was conducted.  

  1. The authors should improve the grammar throughout the manuscript. Many sentences are challenging to understand.

The paper has been proofread by all the authors and updated to reflect grammatical issues raised by the reviewer.

  1. L24: It should be careful to use the term "significant" because it needs to follow the results analyzed by ANOVA.

The term “Significant” was removed from the sentence.

  1. L23-30, L543-562: Very difficult to digest and maybe need a comprehensive graph to present the mechanism and how they are involved in the flavonoid pathway and then contribute to biotic stress resistance.

The authors feel that there are already many graphs and charts and are afraid that adding more might confuse the readers.

Logic and conclusive comments about contribution to biotic stress from our current data is somewhat premature and will be discussed in our next publication.

  1. All descriptions and conclusions according to pure bioinformatics should be more conservative.

The authors agree with the reviewer and thus we have toned down descriptions and conclusions to reflect this point.

  1. M&M: This should be improved because the current version is difficult to follow, maybe a flowchart will be helpful to understand the entire experimental design.

We have followed a standard format of M&M in many IJMS articles reporting closely related experiments. Thus, we need more specific suggestion from the editor if it is necessary.

Reviewer 3 Report

The manuscript was designed and executed well.

Minor comments:

1. The Figure 2 is not clear. Maybe redraw and highlight the specific areas.

2. On Table 2: any NADP+/NADPH ratio?

3. On Figure 5: any explanation to Quercetin0 uM, why at higher concentration , V goes down?

4. On Figure 6: the color of DHK and DHQ is too similar. 

Author Response

Reviewer 3

  1. The Figure 2 is not clear. Maybe redraw and highlight the specific areas.

In the revised version, Figure 2 has been updated to reflect a clearer image of the specified areas.

  1. On Table 2: any NADP+/NADPH ratio?

The authors agree that testing the binding affinity of NADP+/NADPH would be informative data, however using NADPH in ITC measurement will produce a high amount of reaction heat from the hydride transfer reaction and will mask the binding energy that is much lower energy.

  1. On Figure 5: any explanation to Quercetin0 uM, why at higher concentration , V goes down?

We believe that there is either substrate or product inhibition observed at higher concentrations.  The corresponding sentence in the results section has been modified to explain clearly as follow.

“As observed previously, at higher concentrations product inhibition was observed.”

  1. On Figure 6: the color of DHK and DHQ is too similar. 

In the revised version, the color of DHQ has been altered to make it more discernable from DHK.

Round 2

Reviewer 2 Report

The manuscript has been revised and I don't have further questions.

Reviewer 3 Report

No more comments